# A neurodegeneration checkpoint mediated by REST protects against the onset of Alzheimer's disease

Liviu Aron[1,5], Chenxi Qiu[1,5], Zhen Kai Ngian[1], Marianna Liang[1], Derek Drake[1], Jaejoon Choi[1], Marty A. Fernandez[2], Perle Roche[1], Emma L. Bunting[1], Ella K. Lacey [1], Sara E. Hamplova[1], Monlan Yuan[1], Michael S. Wolfe [2], David A. Bennett[3], Eunjung A. Lee[4] & Bruce A. Yankner [1]✉

Many aging individuals accumulate the pathology of Alzheimer's disease (AD) without evidence of cognitive decline. Here we describe an integrated neurodegeneration checkpoint response to early pathological changes that restricts further disease progression and preserves cognitive function. Checkpoint activation is mediated by the REST transcriptional repressor, which is induced in cognitively-intact aging humans and AD mouse models at the onset of amyloid β-protein (Aβ) deposition and tau accumulation. REST induction is mediated by the unfolded protein response together with β-catenin signaling. A consequence of this response is the targeting of REST to genes involved in key pathogenic pathways, resulting in downregulation of gamma secretase, tau kinases, and pro-apoptotic proteins. Deletion of *REST* in the 3xTg and J20 AD mouse models accelerates Aβ deposition and the accumulation of misfolded and phosphorylated tau, leading to neurodegeneration and cognitive decline. Conversely, viral-mediated overexpression of *REST* in the hippocampus suppresses Aβ and tau pathology. Thus, REST mediates a neurodegeneration checkpoint response with multiple molecular targets that may protect against the onset of AD.

A substantial proportion of the aging population can develop the amyloid β-protein (Aβ) and tau pathology of Alzheimer's disease (AD) while remaining asymptomatic[1]. Furthermore, aging individuals with similar levels of AD pathology can have divergent cognitive trajectories; some develop AD, while others remain unimpaired[1–6]. These observations suggest that a subset of the aging population is resilient to AD pathology, and is able to delay or prevent cognitive decline. The mechanisms that underlie neuronal stress resistance and resilience to pathology are largely unknown[7]. Correlative neuropathology and imaging studies have identified several candidate mechanisms of cognitive resilience during aging, such as cellular and synaptic

structural and functional changes[7–9]. However, no clear molecular mechanism for resistance to AD pathology and preservation of cognitive function during aging has emerged.

To gain insight into regulatory mechanisms that protect against the onset of cognitive decline and AD, we characterized the cellular response to early stages of Aβ deposition and abnormal tau accumulation in aging humans and AD mouse models. We uncover a robust activation of the transcriptional repressor REST in association with initial Aβ and tau accumulation in aging humans who maintain cognitive function, and in AD mouse models at early stages of pathology. REST induction is mediated by the unfolded protein response (UPR) and β-catenin

[1]Department of Genetics, Harvard Medical School, Boston, MA 02115, USA. [2]Center for Neurologic Diseases, Brigham and Women's Hospital, Boston, MA 02115, USA. [3]Rush Alzheimer's Disease Center, Rush University Medical Center, Chicago IL60612, USA. [4]Division of Genetics and Genomics, Boston Children's Hospital, Boston, MA 02115, USA. [5]These authors contributed equally: Liviu Aron, Chenxi Qiu. ✉e-mail: bruce_yankner@hms.harvard.edu

signaling. Failure to activate REST in aging humans with AD pathology is a strong predictor of dementia. To determine whether the REST-mediated transcriptional response functions as a checkpoint on neurodegeneration, *REST* was genetically inactivated in excitatory neurons in AD mouse models. Loss of REST markedly accelerates Aβ deposition and abnormal tau accumulation. Moreover, deletion of only a single *REST* allele accelerates neurodegeneration and memory loss. We further show that the REST repressor directly targets multiple gene networks that are involved in the onset and progression of AD, including genes that mediate Aβ generation, tau phosphorylation, and apoptotic signaling. In addition, viral-mediated delivery of the *REST* gene in the hippocampus of AD mice potently suppresses Aβ and tau pathology. These findings suggest that REST mediates an integrated neurodegeneration checkpoint response that restricts the progression of AD pathology, maintains neuronal viability, and preserves cognitive function during aging.

## Results

### REST and the onset of AD pathology in cognitively-intact aging humans and AD mouse models

To explore the role of REST in the aging brain, we examined nuclear REST levels in prefrontal cortical (PFC) neurons in aging individuals with no cognitive impairment (NCI) or clinically diagnosed AD from the ROSMAP cohort (see Methods). The NCI and AD cases were further stratified based on a global composite of plaque and tangle scores into 4 stages—no pathology, early pathology, mid pathology, and late pathology (Supplementary Fig. 1a). These stages were also significantly different when assessed by CERAD, Braak and NIA-Reagan scores (Supplementary Fig. 1a). The mean age at death and post-mortem interval were similar across groups with 39% males and 61% females. Independent analysis did not show a significant effect of gender on the parameters described below (Supplementary Data 1). The majority (76%) of aging individuals in the NCI group showed AD-type pathology (Supplementary Fig. 1b).

Immunofluorescence microscopy showed markedly elevated nuclear REST in NCI cases with early AD-type pathology, which

overlapped the early tau pathology epitope pSer202 tau[10–12] (antibody CP13; Fig. 1a). The REST antibody was previously validated in human brain sections and by REST overexpression and shRNA-mediated knockdown[13] (Methods). Moreover, preincubation of the REST antibody with the antigenic peptide abolished REST immunoreactivity (Supplementary Fig. 1c). In contrast to the cogitively intact NCI cases, REST was largely depleted in AD cases (Fig. 1a). Quantification of REST expression in MAP2-positive cortical neurons showed that nuclear REST was significantly elevated in neurons of NCI cases with early and mid-AD pathology relative to NCI cases with no pathology (Fig. 1b, Supplementary Fig. 1d). In contrast, nuclear REST levels were significantly reduced in AD cases relative to NCI cases with similar levels of pathology, and was not upregulated in AD at any stage of pathology (Fig. 1b, c). When all AD and NCI cases were compared, the mean nuclear REST level was significantly reduced in AD ($P = 3 \times 10^{-14}$) (Fig. 1c). Furthermore, the data distribution in Fig. 1c identified 2 non-overlapping populations: high REST expression (>148 a.u.), which was exclusively detected in cognitively-intact NCI cases, and low REST expression (<35 a.u.), which was exclusively detected in AD cases. These results suggest that REST is induced in cognitively-intact aging individuals with the onset of AD pathology, and that REST induction is absent in these neuronal populations in AD.

We next asked whether REST induction is recapitulated in the 3xTg AD mouse model that expresses human *APP*$^{Swe}$ and *tau*$^{P301L}$ mutant transgenes, and a *PS1* knock-in mutation[14]. The accumulation of early tau pathology (pSer202 tau and misfolded tau) in 11-month old 3xTg mice was associated with REST induction in neurons of both the cortex and the hippocampus (Fig. 2a–c and Supplementary Fig. 2a, b). In contrast, the subsequent appearance of tangle-like tau pathology (positive for the late-stage tau marker pSer396 tau) in older 3xTg mice was associated with loss of REST expression (Fig. 2d–f).

We next asked whether REST can be induced at the onset of Aβ deposition independently of tau. J-20 mice that carry the human *APP*$^{Swe/Ind}$ transgene[15] exhibit age-dependent Aβ accumulation without significant accumulation of tau. REST expression was analyzed in J20

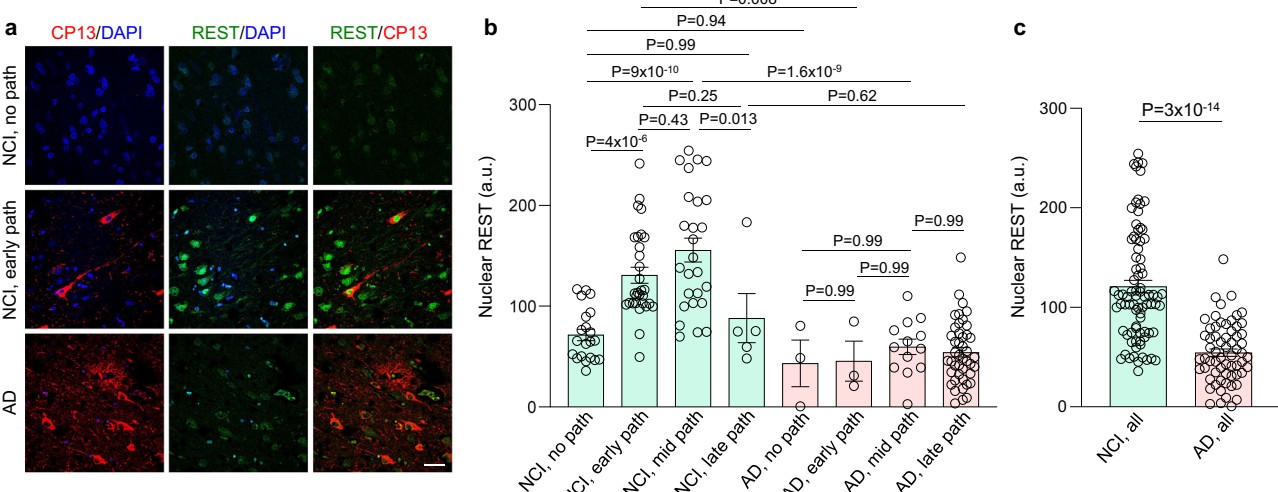

**Fig. 1 | REST and the onset of AD pathology. a** Induction of REST in neurons with early AD pathology. Labeling of REST (green), the early tau pathology marker pSer202-tau (antibody CP13, red) and DNA (DAPI, blue) in the aging human prefrontal cortex shows increased nuclear REST in an NCI case with early AD-type pathology relative to no pathology, and reduced REST levels in AD. CP13 labeling (red) shows broad accumulation of phosphorylated tau in neurites, and in some neuronal cell somas, in NCI cases with early pathology (middle panel), and a strong increase in tau accumulation in neurites and neuronal cell bodies in AD (lower panel). Scale bar, 25 μm. **b** Quantification of nuclear REST levels in pyramidal neurons of the prefrontal cortex in NCI cases with no pathology (n = 21), early pathology (n = 30),

mid pathology (n = 26) and late pathology (n = 5), as well as in AD cases with no pathology (n = 3), early pathology (n = 3), mid pathology (n = 13) and late pathology (n = 44). Neurons were identified by co-labeling with the neuron marker MAP2 (see Supplementary Fig. 1d). See Supplementary Fig. 1b for the relative distribution of each level of pathology in NCI and AD cases. *P*-values were generated by two-way ANOVA with Tukey's post-hoc test. The interaction between cognitive status and pathology stage was significant: F(3, 136) = 3.68, *P* = 0.013). **c** Reduced nuclear REST in AD. The mean nuclear REST level is significantly reduced in AD (n = 63) relative to NCI cases (n = 82). Individual values and the mean ± S.E.M are shown. *P* < 10$^{-12}$ by two-tailed unpaired *t*-test. Source data are provided as a Source Data file.

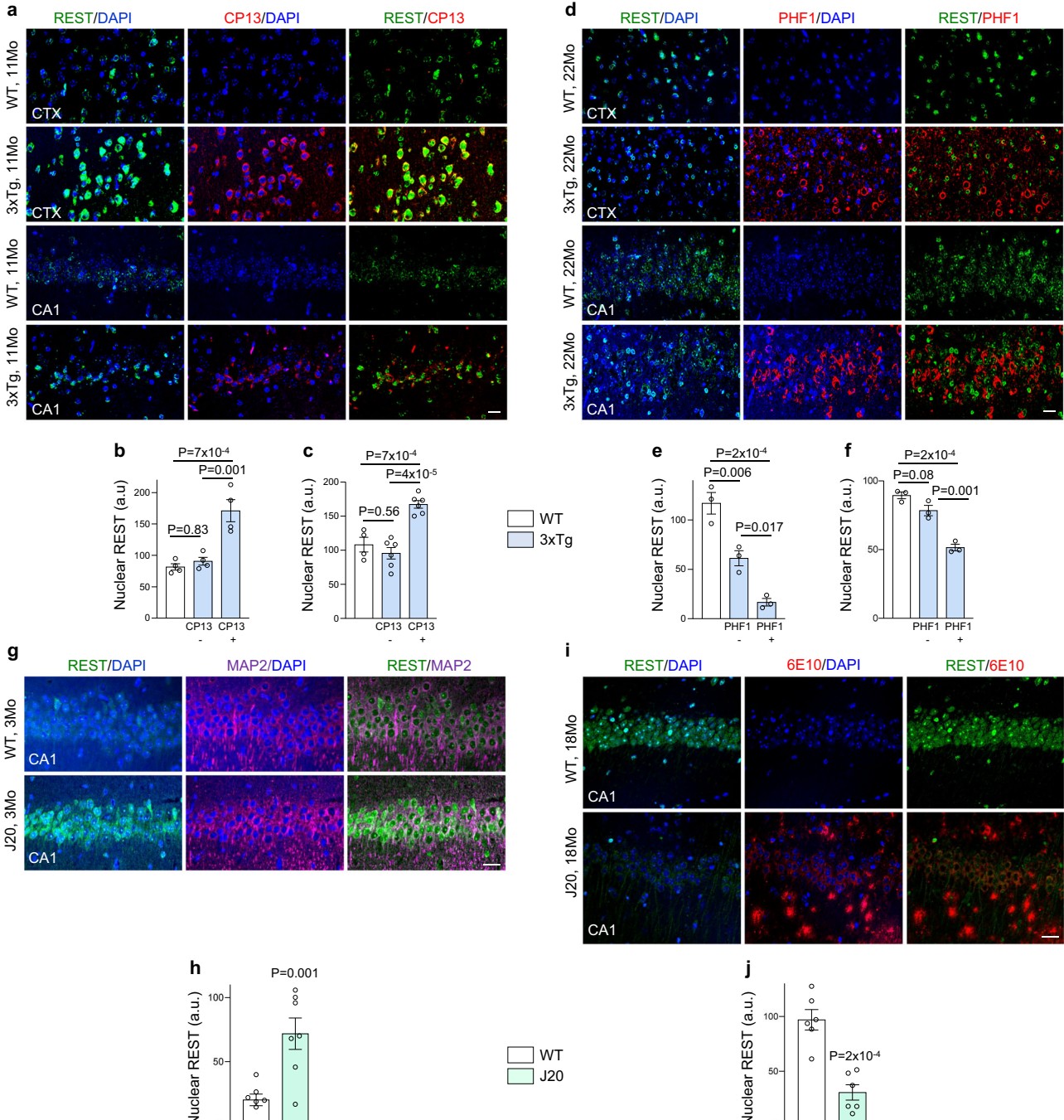

**Fig. 2 | REST induction in AD mice with early pathology. a–c** REST induction in neurons that accumulate early tau pathology. **a** Immunolabeling of REST (green) the marker of early tau pathology, phospho-Ser202 tau (antibody CP13, red) and DNA (DAPI, blue) in the cortex (CTX) and the CA1 region of the hippocampus, shows elevated nuclear REST levels in neurons with accumulation of pSer202 tau in 11-month-old 3xTg mice **b, c** Quantification of nuclear REST levels in 11-month-old WT ($n = 4$) and 3xTg ($n = 4$) cortex (**b**) and 11-month-old WT ($n = 4$) and 3xTg ($n = 6$) hippocampus (**c**). **d–f** Loss of REST in aged 3xTg mice with advanced tau pathology. **d** Immunolabeling of REST (green), phospho-Ser396 tau, a marker of late, fibrillary, tau pathology (antibody PHF1, red) and DNA (DAPI; blue) in the cortex (CTX) and the CA1 region of the hippocampus, shows decreased nuclear REST levels in PHF1-positive neurons in 22-month-old 3xTg mice. Quantification of nuclear REST levels in the cortex (**e**) and hippocampus CA1 sector (**f**) of 22-month-old WT ($n = 3$) and 3xTg ($n = 3$) mice. **g, h** REST induction in J20 mice with early Aβ pathology. **g** Immunolabeling of REST (green), the neuron marker MAP2 (magenta) and nuclei (DAPI, blue) in the CA1 region of the hippocampus, shows elevated nuclear REST levels in 3-month-old J20 mice. **h** Quantification of nuclear REST levels in 3-month-old WT ($n = 7$) and J20 ($n = 7$) mice. **i, j** Loss of REST in aged J20 mice with advanced Aβ plaque pathology. **i** Immunolabeling of REST (green), Aβ (red) and DNA (DAPI, blue) in the CA1 region of the hippocampus, shows decreased nuclear REST levels in 18-month-old J20 mice. **j** Quantification of nuclear REST levels in 18-month-old WT ($n = 6$) and J20 ($n = 6$) mice. **b, c, e, f, h, j** Individual values (representing the average mean fluorescence intensities/mouse) as well as the mean ± S.E.M are shown. *P*-values were generated by one-way ANOVA with Tukey's post-hoc test (**b, c, e, f**) or two-tailed unpaired *t*-test (**h, j**). a.u.-arbitrary units. Scale bars, 25 μm. Source data are provided as a Source Data file.

mice at 3 months of age, before the onset of Aβ plaque pathology (Supplementary Fig. 2c). This showed a robust increase in nuclear REST in hippocampal MAP2-positive neurons relative to age-matched wild-type mice (Fig. 2g, h). In contrast, REST was significantly depleted in older J-20 mice that display widespread Aβ plaque pathology (Fig. 2i, j). Thus, REST is induced during early stages of Aβ and tau accumulation, and is downregulated with progression to later stage pathology and cognitive decline.

## REST induction by synergistic activation of the unfolded protein response and β-catenin signaling

We next explored the mechanism of REST induction in response to early AD pathology. We have previously shown that nuclear β-catenin is elevated in the aging human prefrontal cortex and Wnt/β-catenin signaling induces REST expression in SH-SY5Y neuroblastoma cells[13]. This prompted us to ask whether β-catenin signaling might regulate REST expression during early stages of amyloid and tau accumulation. Examination of the brains of cognitively intact aged individuals with early AD pathology showed that nuclear REST and β-catenin were both elevated and co-localized in the nucleus of neurons in prefrontal cortex (Supplementary Fig. 3a, b). Furthermore, nuclear REST and β-catenin were elevated and colocalized in hippocampal neurons of 3xTg mice with early pathology (Fig. 3a, b). Thus, coordinate induction of nuclear REST and β-catenin is associated with early AD pathology.

To assess the contribution of β-catenin signaling to REST induction, primary cortical neuronal cultures were established from 3xTg mouse embryos. REST was strongly induced in cultured cortical neurons upon accumulation of phospho-tau, but not in wild-type neurons with much lower levels of phospho-tau (Supplementary Fig. 3c, d). 3xTg neuronal cultures were then treated with specific inhibitors of Wnt/β-catenin signaling, including the Wnt antagonist Dickkopf (DKK-1), a tankyrase inhibitor that promotes β-catenin degradation[16] and a selective inhibitor of β-catenin-mediated transcription (ICG-001)[17]. Each of the 3 inhibitors partially abrogated REST induction in 3xTg neurons (Fig. 3c, d). These results suggest that β-catenin signaling contributes to REST induction associated with phospho-tau accumulation.

How does early AD-type pathology lead to the activation of β-catenin and REST? Aβ and tau accumulation have been found to activate the unfolded protein response (UPR) at early stages of AD and in AD mouse models[18]. In addition, the UPR can activate β-catenin signaling in mouse embryonic stem cells[19]. Examination of NCI cases with early AD pathology, as well as 3xTg mice, showed a close correlation between BiP/GRP78, a marker of the UPR, and induction of REST and β-catenin (Fig. 3a, b; Supplementary Fig. 3a, b).

The role of the UPR was examined by treating wild-type neuronal cultures with the classic UPR activator thapsigargin (TG). TG potentiated the induction of nuclear β-catenin following activation of β-catenin signaling by the drugs CHIR99021 or lithium chloride, which are inhibitors of GSK3β (Supplementary Fig. 3e, f). Furthermore, the combination of UPR and β-catenin activation dramatically elevated nuclear REST levels relative to either alone (Fig. 3e, f).

The UPR is activated in neurons with early accumulation of tau through phosphorylation of eIF2α by the PERK kinase in aging human neurons[20]. To explore the role of PERK in REST induction, 3xTg primary cortical neuronal cultures were treated with the PERK kinase inhibitor GSK2606414. This significantly reduced nuclear REST (Fig. 3c, d) suggesting a role for PERK and the UPR in the induction of REST. Taken together, these results suggest that REST is activated by a synergistic effect of the UPR and β-catenin signaling.

## REST target genes and mechanisms of neuroprotection

To explore the mechanisms of neuroprotection associated with the neurodegeneration checkpoint, we defined genes that are targeted by

REST in response to the onset of AD-type pathology in 3xTg mice. REST ChIP-seq analysis of the cortex was performed with a validated C-terminal REST antibody[21] in 11-month-old 3xTg and littermate control mice (4 biological replicates, each comprised of 4 pooled cortices, per genotype; Supplementary Data 2). To assess specificity, DNA sequence analysis of the ChIP-seq peaks was performed which showed that the known and de novo REST RE1 binding motifs were highly enriched (Fig. 4a). REST binding sites were most enriched in gene promoters within 1 kb of the transcription start site (Fig. 4b and Supplementary Fig. 4a).

ChIP-seq resolved overlapping and unique REST targets in the WT and 3xTg mouse cortex (Supplementary Data 3 and 4). A subset of REST target genomic regions (peaks) was shared between WT and 3xTg mice (Fig. 4a and Supplementary Data 4). The shared REST target genes were highly enriched in gene ontology (GO) groups corresponding to nervous system development, cell signaling, intracellular transport and metabolic processes (Supplementary Fig. 4b, Supplementary Data 4). Analysis of differentially bound genes in WT and 3xTg identified 507 peaks (635 genes) with significantly higher REST occupancy, and no peaks with significantly decreased REST occupancy, in the 3xTg cortex (Supplementary Data 5). REST targets induced in 3xTg were enriched in GO groups related to cellular metabolism, vesicle mediated transport in synapses, cytoskeletal organization, cellular stress response, apoptosis and the cell cycle (Fig. 4c, Supplementary Data 5). Induced REST targets included genes that may contribute to AD pathogenesis, such as the tau kinases cyclin-dependent kinase (dk5) and glycogen synthase kinase 3β (Gsk3b), and proapoptotic genes including Daxx, Foxo3, Gadd45a and Casp9 (Fig. 4d and Supplementary Data 5).

ChIP-qPCR confirmed significantly elevated REST binding to the tau kinase genes Cdk5 and Gsk3b, and the inflammation-related proapoptotic gene Daxx in 3xTg mice (Fig. 4e). No significant signal was observed in control regions 10 kb downstream from the REST binding sites, or by performing the ChIP with a non-specific IgG control antibody (Fig. 4e). Furthermore, ChIP-qPCR confirmed two loci with strong REST binding by ChIP-seq (Tle3 and Cacng2), and one region devoid of REST binding (Untr6) (Supplementary Fig. 4c, d). Thus, REST binding to genes that mediate tau phosphorylation and cell death is increased in 3xTg mice with early AD-type pathology.

## Loss of REST induces the tau kinases CDK5 and GSK3β

To determine whether loss of REST expression, as observed in AD, leads to derepression of key pathogenic genes implicated by ChIP-seq, mice were generated with either a conditional REST deletion in neurons or a heterozygous REST deletion in all cells. A conditional postnatal (P19) REST deletion in CA1 glutamatergic hippocampal neurons was generated using T29-1 CamKIIα-Cre[22], or in glutamatergic neurons of the entire cortex and hippocampus using a CamKIIα-Cre line with broad Cre expression[23]. Mice carrying Cre and REST^{tx/tx} alleles[24] are referred to as REST cKO (conditional knockout), and mice carrying Cre and REST^{tx/+} are referred to as REST cHET (conditional heterozygous knockout). REST cKO mice showed loss of REST expression in neurons in situ (Supplementary Fig. 5a, e), and significant reduction in REST mRNA and protein levels in bulk brain tissue, which was partial due to deletion in only excitatory neurons (Supplementary Fig. 5b–d). In addition, crossing of REST^{tx/tx} to mice with a ubiquitously expressed CMV-Cre transgene provided evidence that the REST^{rec} generated by Cre-mediated recombination of REST^{tx} is a strong loss-of-function allele (Supplementary Fig. 5f, g).

The REST conditional deletion did not affect the levels of APP or tau transgene expression when crossed to 3xTg or J20 AD mouse lines (Supplementary Fig. 5h–l). To independently validate the role of REST, we utilized a REST gene trap (REST^{GT}) allele in which a β-geo cassette was inserted in the REST intron between non-coding exon 1a-c and the first coding exon, exon 2, leading to a complete REST null allele[25].

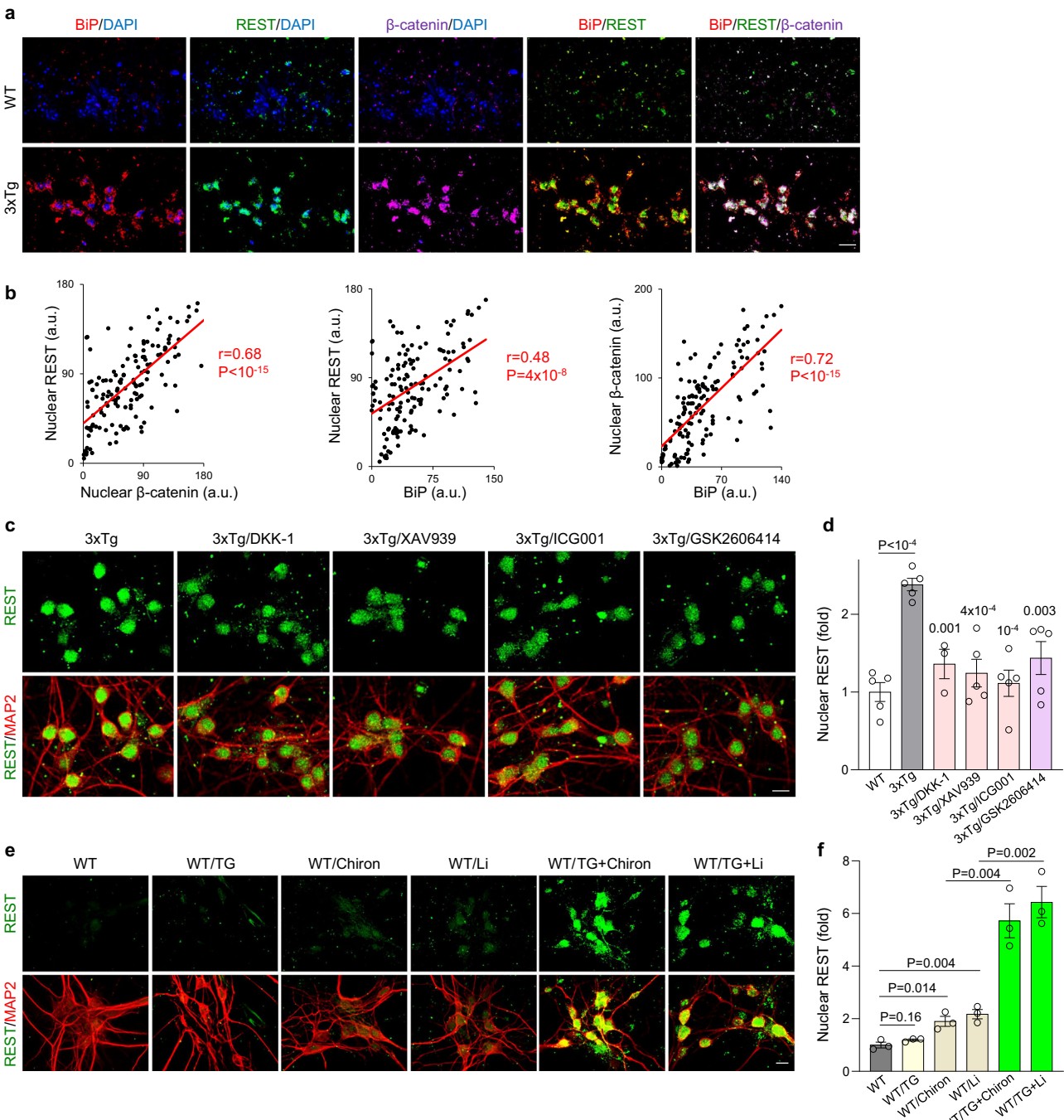

**Fig. 3 | Role of the UPR and Wnt/β-catenin signaling in nuclear REST induction.**
**a** Immunolabeling of UPR activation (marker BiP/GRP78, red), REST (green), β-catenin (magenta) and DNA (DAPI, blue) in 11-month-old 3xTg and WT mice shows coordinate upregulation of BiP, nuclear REST and nuclear β-catenin expression in 3xTg mice. **b** Correlation between nuclear β-catenin and nuclear REST (left graph), BiP and nuclear REST (middle graph) and BiP and nuclear β-catenin (right graph) levels in the hippocampus CA1 region of 11-month-old 3xTg mice. Shown are the mean fluorescence intensity values for nuclear REST in individual CA1 neurons from $n = 3$ 3xTg mice. a.u.- arbitrary units. The Pearson correlation coefficient (r) and P-value are shown. **c** Immunolabeling of REST (green) and neuron marker MAP2 (red) in 3xTg primary cortical neurons (PCNs) shows nuclear REST in neurons treated with vehicle and decreased nuclear REST in 3xTg neurons after a 24-h treatment with the Wnt/β-catenin inhibitors Dickkopf 1 (DKK1), XAV939 and ICG001, or the PERK inhibitor GSK2606414. **d** Quantification of average nuclear REST levels in WT PCNs, as well as 3xTg PCNs treated with vehicle or the individual drugs. The P-

values for planned comparisons (using two-tailed unpaired t-test) between each treated group and 3xTg treated with vehicle are shown; for WT vs 3xTg/vehicle, $P < 10^{-4}$. **e** Immunolabeling of REST (green) and neuron marker MAP2 (red) in WT primary cortical neurons (PCNs) shows low nuclear REST in WT neurons treated with vehicle or the UPR inducer thapsigargin (TG), mildly increased nuclear REST expression after treatment with the GSK3β inhibitors CHIR99021 (Chiron) and lithium chloride (Li), and strongly increased nuclear REST levels after treatment with a combination of TG and Chiron, or a combination of TG and Li. **f** Quantification of average nuclear REST levels. P-values for pre-planned comparisons, using two-tailed unpaired t-tests, are indicated. **d**, **f** Nuclear REST levels are expressed as mean fluorescence intensity/nucleus (in arbitrary units, a.u.) for $n = 5$ (all drugs except DKK-1) or $n = 3$ (**f**, and DKK-1 in panel **e**) independent experiments. Individual values and the mean ± S.E.M are shown. Scale bars, 25 μm. Source data are provided as a Source Data file.

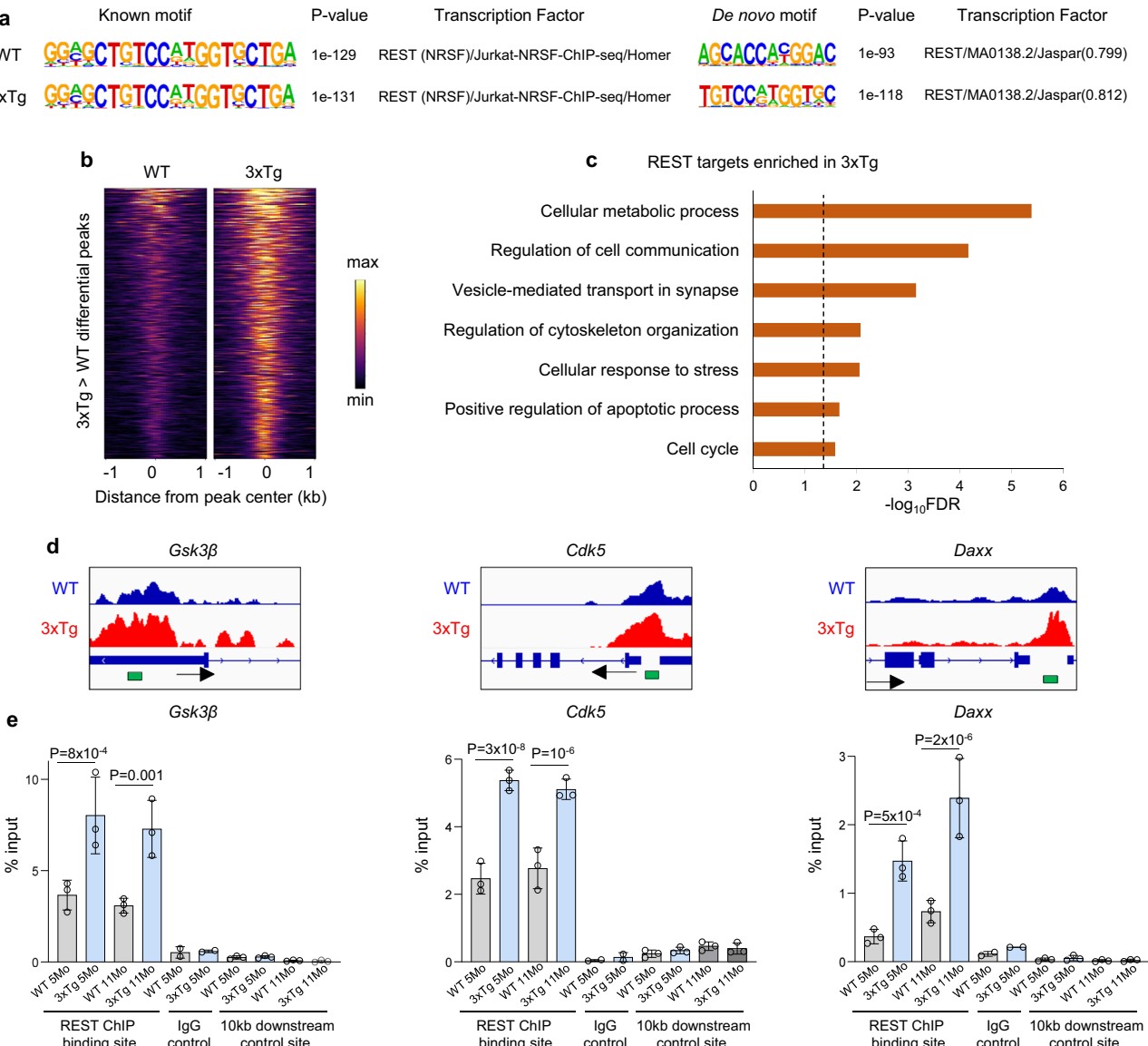

**Fig. 4 | REST targets key pathogenic pathways in AD. a** Analysis of known (left panels) and de novo motifs (right panels) shows that the REST-RE1 DNA binding motif is highly enriched in the ChIP-seq peaks in both 3xTg and WT mice. **b** Heat maps showing REST binding to the 507 REST peaks that exhibit significantly higher REST binding in the 3xTg vs. WT cortex. Peak regions are sorted from highest to lowest REST binding levels. **c** Gene ontology analysis of genes located under the peaks that are predominantly bound in 3xTg mice shows that REST targets genes that regulate cellular metabolism, cell communication, stress responses, and apoptotic cell death. Dashed line: FDR = 0.05. **d** Examples of REST binding to selected genomic regions in WT (blue: shown is the average of 4 biological replicates) and 3xTg (red: average of 4 biological replicates) mouse cortex resolved by ChIP-seq. The arrows indicate the transcription start sites. The regions within REST ChIP-seq peaks that were amplified by qPCR are indicated in green. **e** ChIP-qPCR analysis REST binding to peak regions in *Cdk5*, *Gsk3β*, and *Daxx* genes, in 5- and 11-month-old WT and 3xTg cortex. Also shown is binding by a non-specific IgG control antibody, as well as REST binding to control sites located 10 kb downstream from the REST peaks. *n* = 3 mice/group. Individual values and the mean ± S.E.M are shown. *P*-values were generated by one-way ANOVA analyses with Šídák's multiple comparisons test, followed by Bonferroni correction for the simultaneous testing of 6 genomic regions (see Supplementary Data 1 for all Bonferroni-adjusted *P*-values). Source data are provided as a Source Data file.

$REST^{GT/GT}$ mice die at embryonic day 9.5–11.5, but $REST^{GT/+}$ mice are viable[25], and have been maintained for up to 29 months. We generated mice carrying a single REST gene trap ($REST^{GT}$) null allele in AD backgrounds. The *GT* allele did not alter the levels of holo-APP or total tau (Supplementary Fig. 5m–o).

The expression of the major tau kinases CDK5 and GSK3β was assessed in 3xTg mice with *REST* deletion, using both conditional and gene trap REST alleles. REST inactivation significantly increased the expression of both CDK5 (Fig. 5a–d) and GSK3β (Fig. 5e–h) in neurons of the hippocampus and cortex. This was confirmed by Western blot analysis, which showed elevated CDK5 and GSK3β levels in aged 3xTg;GT relative to 3xTg mice (Fig. 5i, j).

We next asked whether REST induction in aged individuals with no cognitive impairment (NCI) but with early AD pathology is associated with repression of CDK5 and GSK3β. Double immunofluorescent labeling for REST and either CDK5 or GSK3β showed that nuclear REST levels are inversely correlated with CDK5 and GSK3β expression in prefrontal cortical neurons in these cases (Fig. 5k–n). These results suggest that loss of REST leads to derepression of the tau kinases CDK5 and GSK3β.

**REST suppresses gamma secretase and Aβ generation**
Previous studies have shown that REST can suppress the expression of presenilins in *C.elegans* and in the aging human brain[13,26]. To determine

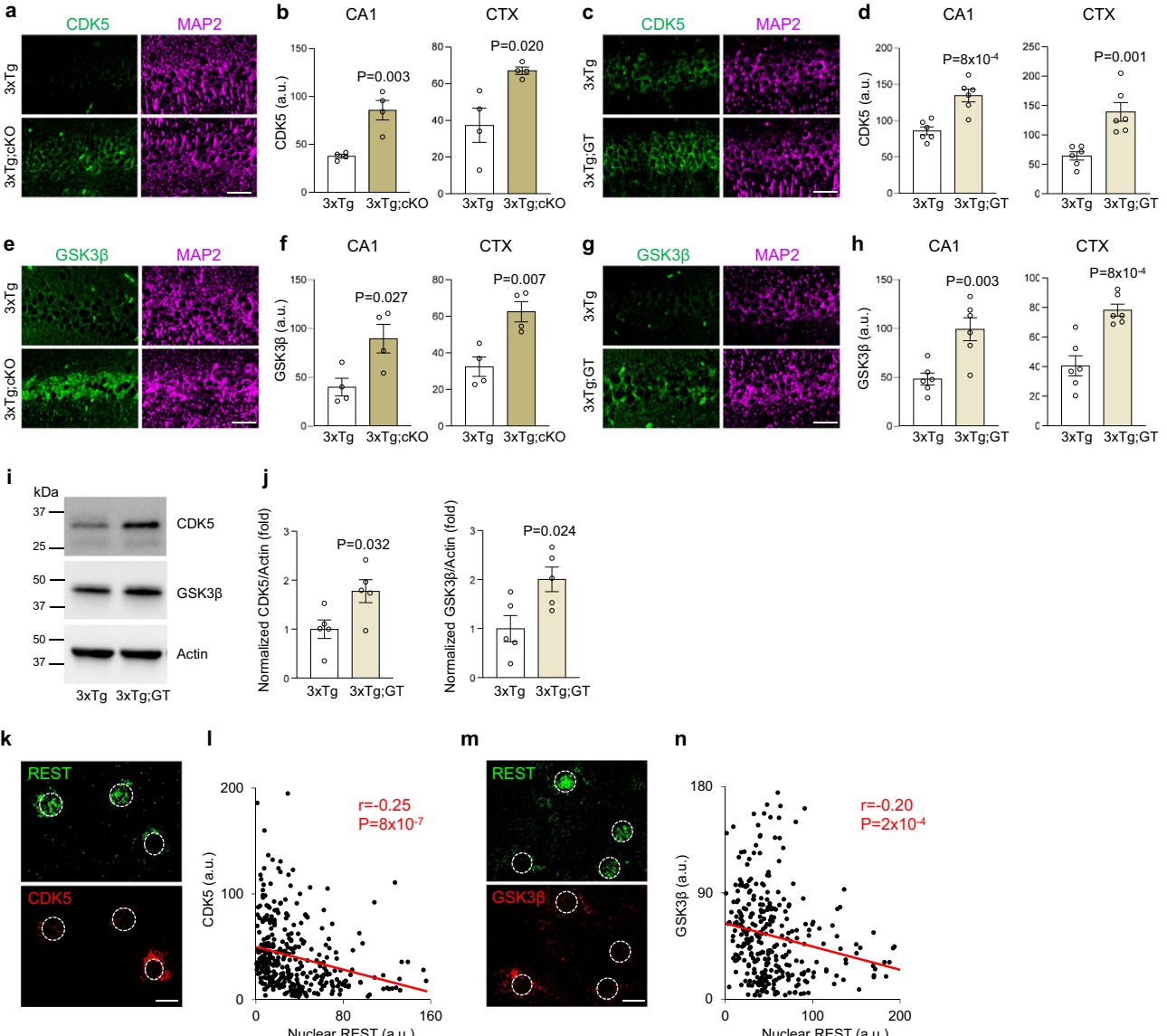

**Fig. 5 | REST suppresses the tau kinases CDK5 and GSK3β. a, b** Loss of REST in excitatory neurons increases CDK5 expression in cortex and hippocampus. **a** Immunolabeling for CDK5 (green) and the neuronal marker MAP2 (magenta) in CA1 neurons of the hippocampus in 9-month-old 3xTg and 3xTg;cKO mice. **b** Quantification of CDK5 immunofluorescence intensity in the hippocampus and cortex of 9-month-old 3xTg ($n = 4$) and 3xTg;cKO ($n = 4$) mice. **c, d** Loss of a single REST allele increases CDK5 expression. **c** Immunolabeling for CDK5 (green) and MAP2 (magenta) in 29-month-old 3xTg and 3xTg;GT (heterozygous REST null) mice. **d** Quantification of CDK5 immunofluorescence intensity in 28–29-month-old 3xTg ($n = 6$) and 3xTg;GT ($n = 6$) mice. **e, f** Loss of REST in excitatory neurons increases GSKβ expression in cortex and hippocampus. **e** Immunolabeling for GSK3β (green) and MAP2 (magenta) in hippocampal CA1 neurons in 9-month-old 3xTg and 3xTg;cKO mice. **f** Quantification of GSK3β immunofluorescence intensity in 9-month-old 3xTg ($n = 4$) and 3xTg;cKO ($n = 4$) mice. **g, h** Loss of a single REST allele increases GSK3β expression. **g** Immunolabeling for GSK3β (green) and MAP2 (magenta) in 29-month-old 3xTg and 3xTg;GT (heterozygous REST null) mice. **h** Quantification of GSK3β immunofluorescence intensity) in 28–29-month-old 3xTg ($n = 6$) and 3xTg;GT ($n = 6$) mice. **i** Western blot analysis of CDK5 and GSK3β levels in the hippocampus of 29-month-old 3xTg and 3xTg;GT mice. **j** Quantification shows CDK5 and GSK3β levels normalized to actin in 28–29-month-old 3xTg ($n = 5$) and 3xTg;GT ($n = 5$) mice. **b**, **d**, **f**, **h**, **j** Individual values and the mean ± S.E.M are shown; P-values are derived from two-tailed unpaired t-tests. **k**, **m** Immunofluorescence labeling of REST (green) and CDK5 (red, **k**) or GSK3β (red, **m**) in a NCI case with early AD pathology shows an inverse relationship between nuclear REST and either CDK5 (**k**) or GSK3β (**m**) total cellular levels in neurons of the prefrontal cortex. **l**, **n** Correlation between nuclear REST levels and total cellular levels of CDK5 (**l**) or GSK3β (**n**) in individual neurons from $n = 3$ NCI cases with early pathology. Shown are the Pearson r correlation coefficients and the P-values. Scale bars, 25 µm. Source data are provided as a Source Data file.

if REST regulates the γ-secretase complex in the setting of early AD pathology, we evaluated the expression of core components of γ-secretase in *REST*-deficient 3xTg mice. Immunofluorescence labeling showed that the expression of presenilin 1 (PS1) was significantly elevated in MAP2-positive cortical and hippocampal neurons of *3xTg;cKO* and *3xTg;GT* relative to *3xTg* mice (Supplementary Fig. 6a–c). In addition, PEN2 and nicastrin expression were elevated in cortical and hippocampal neurons of *3xTg;cKO* and *3xTg;GT* relative to *3xTg* mice

(Supplementary Fig. 6d–i). These results suggest that REST represses the expression of multiple components of the γ-secretase complex in the setting of early AD-type pathology.

To further explore the regulation of γ-secretase, REST expression was reduced in human neuroblastoma SH-SY5Y cells using short hairpin RNAs that have been previously validated[13]. REST knockdown significantly upregulated the γ-secretase genes *PS2, PEN2, NCSTN* and *APH1* (Fig. 6a). Conversely, lentiviral-mediated transduction of human

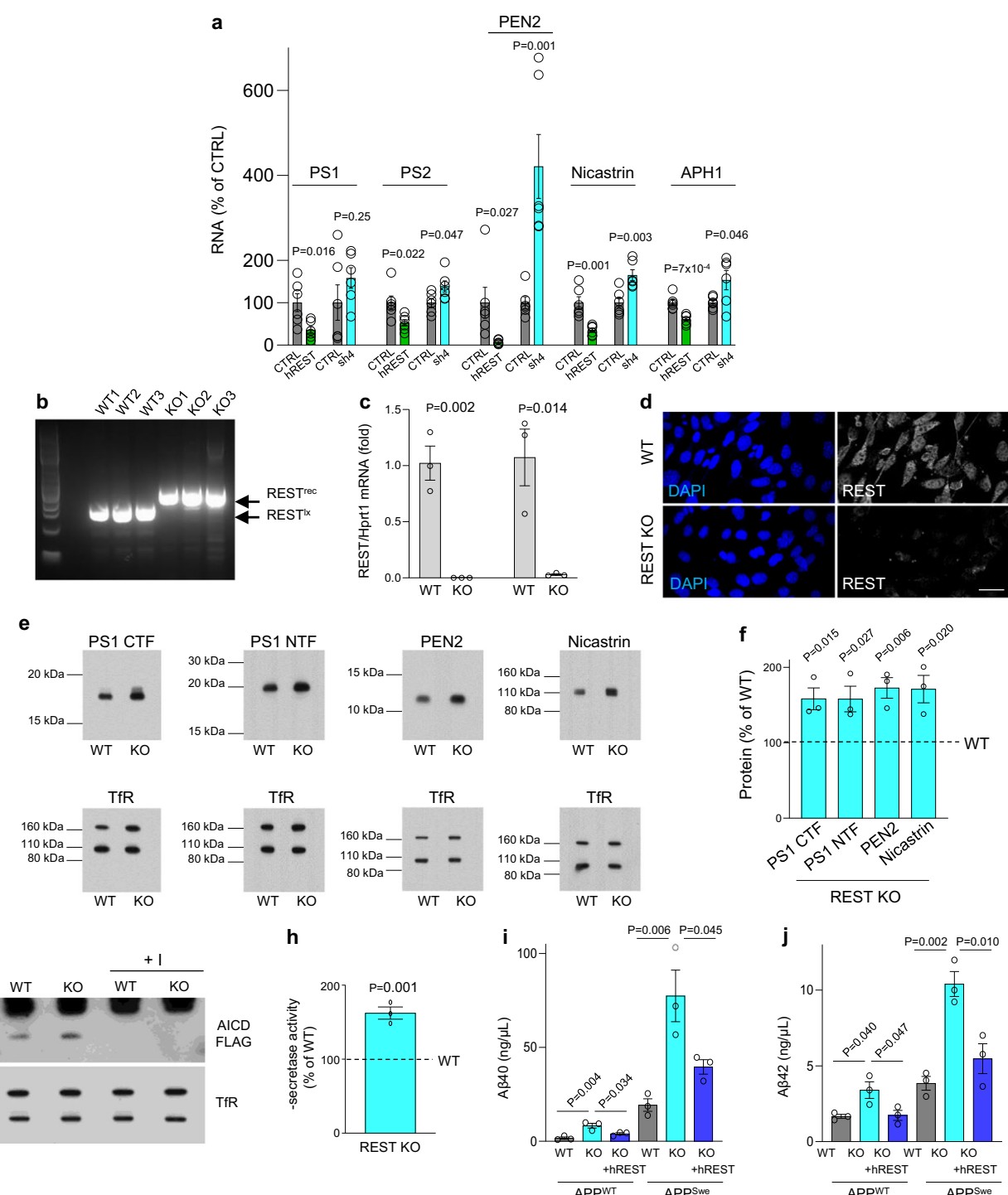

REST cDNA downregulated the expression of every member of the γ-secretase complex (Fig. 6a). We corroborated these findings by inactivating REST in mouse embryonic fibroblasts (MEFs) derived from $REST^{tx/tx}$ embryos by retroviral transduction of Cre recombinase (Fig. 6b–d). REST KO MEFs exhibited increased expression of PS1, PEN2 and nicastrin (Fig. 6e, f).

To directly assess the proteolytic activity of γ-secretase, we utilized Met-C99-FLAG, an epitope-tagged recombinant version of C99, the APP C-terminal fragment generated by β-secretase cleavage. Cleavage of C100-FLAG by γ-secretase generates a FLAG-tagged APP intracellular domain (AICD-FLAG) and Aβ[27]. Solubilized membranes from REST KO MEFs incubated with C100-FLAG generated significantly higher levels of AICD-FLAG compared to membranes from WT MEFs

(Fig. 6g, h), suggesting that REST inactivation increases γ-secretase activity. Moreover, REST inactivation in MEFs significantly increased the generation of Aβ40 and Aβ42 following transfection of either APP^WT or APP^Swe cDNAs (Fig. 6i, j). As expected, APP^Swe expression led to higher Aβ levels than APP^WT (Fig. 6i, j). Finally, lentiviral transduction of human REST in REST-KO MEFs significantly reduced Aβ40 and Aβ42 levels (Fig. 6i, j). Taken together, these results suggest that both mouse and human REST suppress γ-secretase and inhibit the generation of Aβ.

## REST inactivation accelerates Aβ deposition and pathogenic tau accumulation

What is the consequence of loss of REST on the progression of AD-type pathology in the brain? To explore this question, we compared tau

**Fig. 6 | REST suppresses γ-secretase. a** Cultured human SHY-5Y neuroblastoma cells were transduced with recombinant lentiviruses leading to either REST inhibition (short hairpin RNA; sh4) or overexpression (human *REST* cDNA: h*REST*), as previously described[10]. Expression of γ-secretase components was assessed by qRT-PCR. **b** *REST*^{tx/tx} MEF cells were transduced with *Cre* recombinase (generating REST-KO lines 1–3) or a control vector without *Cre* (lines WT 1–3). The *REST* floxed (*REST*^{tx}) and Cre-recombined *REST* (*REST*^{rec}) alleles were detected by PCR. No *REST*^{tx} band was detected in REST-KO cells, suggesting complete *Cre*-mediated recombination. **c** qRT-PCR using two sets of primers spanning the transcript shows loss of *REST* mRNA in REST-KO MEFs. **d** Immunolabeling with anti-REST antibody (REST14[44]; white) shows loss of REST expression in REST-KO MEFs. Nuclei are labeled with DAPI (blue). Scale bar, 40 μm. **e** Western blot analysis of γ-secretase components in WT and REST-KO MEFs. The transferrin receptor (TfR) served as loading control. **f** Quantification of protein levels normalized to TfR shown as percentage expression in REST-KO relative to WT cells (interrupted line: 100%). For PS1, similar results

were seen with antibodies against the N-terminus (NTF; antibody 231f) or C-terminus (CTF; antibody EP2000Y). **g** REST-KO MEFs show elevated γ-secretase enzymatic activity. Solubilized membranes were incubated with Met-C99-FLAG in the presence or absence of a γ-secretase inhibitor (+I), and levels of AICD-FLAG were determined by Western blotting using TfR as a loading control. **h** Quantification of γ-secretase activity in membrane preparations from WT and REST-KO cells. AICD-FLAG levels were normalized to TfR, and shown as percentage expression in REST-KO relative to WT cells (interrupted line: 100%). Loss of REST in REST-KO MEFs leads to elevated Aβ40 (**i**) and Aβ42 (**j**) levels following transfection of hAPP^{WT} or hAPP^{Swe}. Lentiviral transduction of human REST cDNA (hREST) suppresses Aβ production. Individual values and the mean ± S.E.M are shown for *n* = 6 (**a**) or *n* = 3 (**c**, **f**, **h**, **i**, **j**) independent experiments. *P*-values are derived from two-tailed unpaired *t*-tests (**a**, **c**, **f**, **h**) or one-way ANOVA with Tukey's post-hoc test (**i**, **j**). Source data are provided as a Source Data file.

pathology in 17–18 month-old 3xTg mice and *3xTg;cKO* mice with either a CA1-specific or broader forebrain REST deletion in glutamatergic neurons. Deletion of *REST* increased the accumulation of misfolded and phosphorylated tau, as determined by immunolabeling with antibodies MC1 and CP13, respectively (Fig. 7a–c). Moreover, partial loss of REST in *3xTg;cHET* mice was sufficient to significantly augment tau accumulation (Fig. 7b, c). Western blotting with the phospho-tau antibodies PHF1 and AT180 confirmed that REST inactivation elevates the level of phospho-tau species typically associated with neurofibrillary degeneration[10,12] (Fig. 7d, e). Thus, loss of REST augments the accumulation of pathogenic forms of tau.

To assess the early stages of pathogenic tau accumulation, we established primary cortical neuronal cultures from 3xTg mouse embryos. REST was inactivated by lentiviral transduction of Cre recombinase in neuronal cultures derived from *3xTg;REST*^{tx/tx} mice (Supplementary Fig. 7a). *REST* deletion markedly increased the accumulation of misfolded tau immunoreactive with the conformation-specific tau antibody MC1 (Supplementary Fig. 7b, c). Thus, loss of REST accelerates tau misfolding, an early step in the development of neurofibrillary pathology in AD.

We next asked whether REST regulates the deposition of Aβ. To address this question, we examined J20 mice that carry an hAPP^{Swe/lnd} transgene and consequently exhibit robust age-dependent Aβ deposition[15]. Crossing of J20 and REST cKO mice generated *J20;cHET* and *J20;cKO* mice, with partial and complete *REST* deletion, respectively, which did not affect the level of APP transgene expression (Supplementary Fig. 5k, l). Both *J20;cHET* and *J20;cKO* mice exhibited significantly elevated Aβ plaque burden in the hippocampus and cortex by 12 months of age (Fig. 7f, g). Thus, even partial loss of *REST* can markedly accelerate Aβ deposition.

### Accelerated Aβ deposition and tau accumulation in AD mice with a *REST* genetrap allele
To confirm the phenotype of the conditional *REST* deletion mouse model, we examined mice that carry a single *REST*^{GT} allele, leading to heterozygous deletion of *REST* in all cells. We began by examining hemizygous *3xTg* mice that exhibit delayed onset of pathology[14]. Only 1 of 11 aged (27–29 month old) hemizygous 3xTg mice (9%) exhibited cortical or hippocampal Aβ plaque deposition. In contrast, 7 of 9 (77%) hemizygous 3xTg littermates bred to carry a single *REST*^{GT} null allele exhibited amyloid plaque deposition in the hippocampus, and 3 of 9 (33%) exhibited plaques in the cortex. Quantification of Aβ plaque burden showed a significant increase in aged *3xTg;GT* hippocampus, as well as a trend in the cortex Supplementary Fig. 7d, e).

We next examined 27–29 month-old J20 mice carrying a single *REST*^{GT} null allele. Aβ plaque deposition was significantly elevated in the cortex in *J20;GT* mice (Supplementary Fig. 7e). Hippocampal Aβ deposition was abundant in *J20* mice at this age, and was not significantly further elevated in *J20;GT* mice (Supplementary Fig. 7f, g).

Thus, partial REST deletion is sufficient to accelerate Aβ deposition. Partial inactivation of *REST* also led to significantly elevated accumulation of phospho-tau in the hippocampus of 3xTg mice (Supplementary Fig. 7h, i). These results confirm that partial deletion of *REST* accelerates Aβ deposition and tau accumulation in AD mouse models, suggesting that REST can inhibit the progression of AD-type pathology.

### Loss of REST promotes phospho-tau accumulation in a pure APP-transgenic mouse model
The lack of significant tau pathology in mice that carry familial AD mutations in *APP* in the absence of *tau* transgenes suggests that these models might lack an essential component of AD pathogenesis. To explore the role of REST, we asked whether *REST* deletion in a pure APP transgenic mouse model would affect tau. Initially, *J20* and *J20;GT* mice with partial REST deletion were assessed with an antibody to pThr217-tau, an early marker of tau accumulation in individuals with cognitive decline[28–30]. *J20;GT* mice exhibited significantly elevated hippocampal pThr217-tau levels relative to *J20* with a normal complement of REST (Supplementary Fig. 8a, b). Another early marker of tau pathology, pSer202-tau (antibody CP13), was also significantly elevated in aging *J20;GT* mice (Supplementary Fig. 8c, e). Specificity for the phosphorylated epitope was confirmed by phosphatase preincubation (Supplementary Fig. 8d). In addition to early tau markers, the more advanced stage marker pThr231-tau (antibody AT180) was also elevated, and the late stage marker pSer202/Thr205-tau (antibody AT8) showed an elevated trend in *J20;GT* (Supplementary Fig. 8b). The level of total tau did not significantly change (Supplementary Fig. 8a, b). Thus, Aβ deposition gives rise to phospho-tau accumulation upon loss of REST, suggesting that REST might prevent the development of neurofibrillary pathology at the early stages of Aβ deposition.

### REST protects against neurodegeneration
To assess the role of REST as a determinant of neurodegeneration, neuronal viability was determined in aging 27–29 month old *3xTg;GT* mice with a partial deletion of *REST*. *3xTg;GT* mice exhibited significant neuronal loss in the hippocampal CA1 and CA3 subfields, and in the cortex, relative to *3xTg* mice with a normal REST complement (Supplementary Fig. 9a, b). In addition, conditional deletion of REST in excitatory neurons also resulted in significant neuronal loss in the hippocampal CA1 and CA3 subfields in 3xTg;cKO mice (Supplementary Fig. 9c, d). Labeling of neurons for NeuN confirmed significant loss of CA1 neurons in both 3xTg;cKO and 3xTg;cHET mice relative to 3xTg mice (Supplementary Fig. 9e, f). Ongoing neurodegeneration was assessed by terminal deoxynucleotidyl transferase-dUTP nick end labeling (TUNEL), which identified degenerating neurons in *3xTg;GT*, but not *3xTg* or control mice (Fig. 8a, b). Neurons that accumulated pTau were preferentially labeled by TUNEL in *3xTg;GT* mice (Fig. 8a,

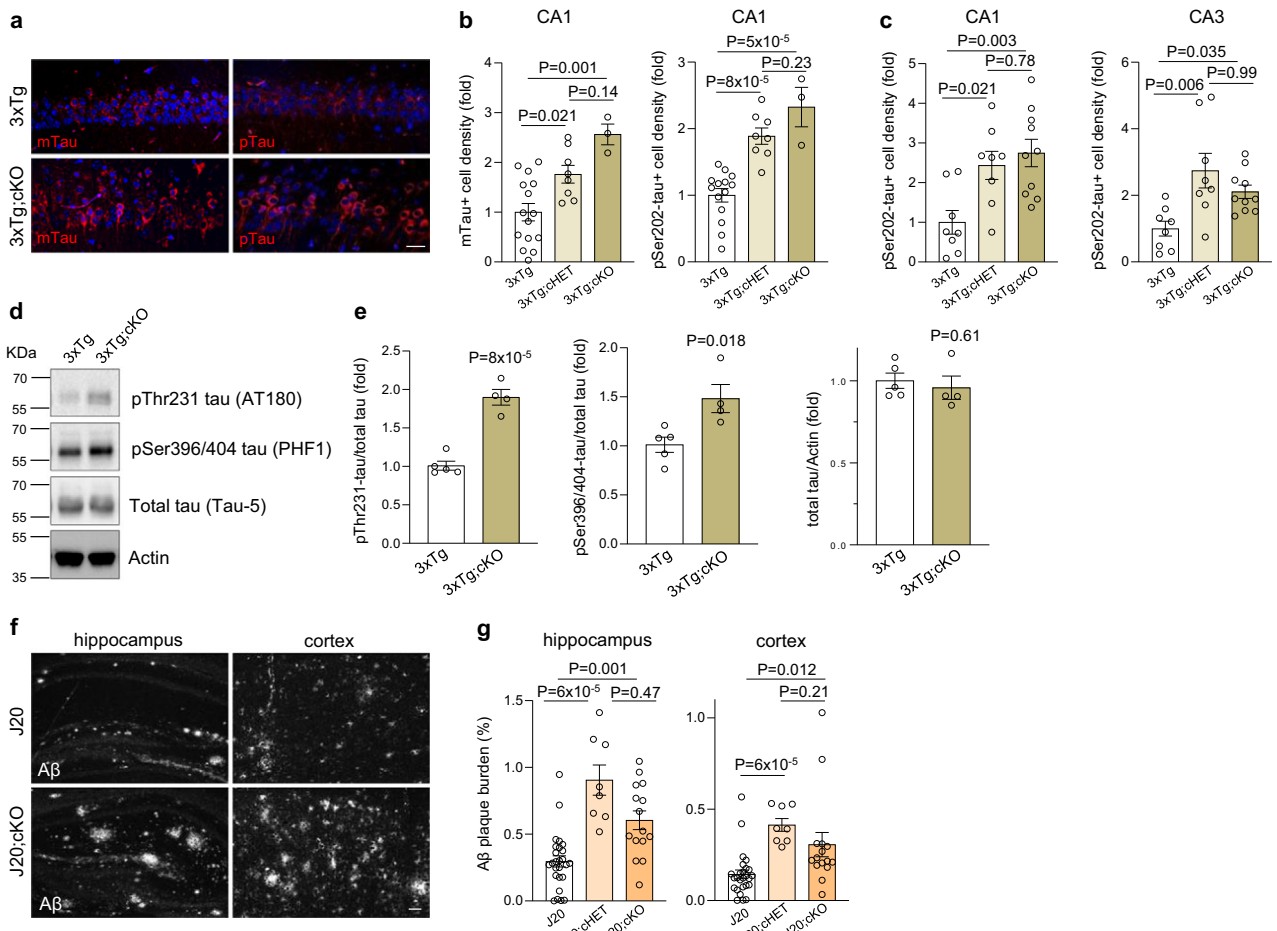

**Fig. 7 | REST suppresses tau accumulation and Aβ deposition in AD mouse models. a**, **b** Increased mTau and pTau accumulation in REST-deficient 3xTg mice. **a** Immunolabeling for conformationally altered tau (antibody MC1, mTau) or pSer202 tau (antibody CP13, right panels, pTau), and DNA (DAPI) in 18-month-old 3xTg and 3xTg;cKO mice with a CA1 neuron-specific REST deletion. **b** Quantification of mTau-positive and pTau-positive neuron density in the hippocampal CA1 sector of 17–18-month-old 3xTg (n = 15), 3xTg;cHET (n = 8) and 3xTg;cKO (n = 3) mice. Scale bar, 25 μm. **c** Quantification of pTau-positive neuron density in 17–18-month-old 3xTg (n = 8), 3xTg;cHET (n = 8) and 3xTg;cKO (n = 10) mice generated with a second Cre line that gives rise to conditional forebrain deletion of REST in glutamatergic neurons. **d**, **e** REST deletion induces multiple phospho-tau epitopes associated with neurofibrillary pathology. **d** Western blot analysis of pThr231-tau

(antibody AT180) and pSer396-tau (antibody PHF1), as well as total tau (antibody tau-5) and actin, in the hippocampus of 9-month-old 3xTg (n = 5) and 3xTg;cKO (n = 4) mice. **e** Quantification of pThr231-tau:total tau, pSer396-tau:total tau and total tau:Actin ratios in 9-month-old 3xTg and 3xTg;cKO mice. **f**, **g** Increased Aβ plaque deposition in aged REST-deficient J20 mice. **f** Immunolabeling of Aβ (white) in 14-month-old J20 and J20;cKO hippocampus and cortex. **g** Quantification of Aβ plaque burden in the hippocampus (left bar graph) and the cortex (right bar graph) in 12–14-month-old J20 (n = 26), J20;cHET (n = 8), and J20;cKO (n = 15) mice. Scale bar, 200 μm. **b**, **c**, **e**, **g** Individual values as well as the mean ± S.E.M are shown. *P*-values were generated by one-way ANOVA with Tukey's post-hoc test (**b**, **c**), Kruskal–Wallis one-way ANOVA with Dunn's post-hoc test (**g**) or two-tailed unpaired *t*-test (**e**). Source data are provided as a Source Data file.

pTau/TUNEL). These results suggest that loss of REST accelerates neurodegeneration in neurons that accumulate pTau.

We next assessed the effects of REST on neuronal survival in aged 27–29 month old J20 mice that develop amyloid deposits in the absence of tau pathology. There was significantly elevated neuronal loss in J20;GT relative to J20 mice in the hippocampal CA1 and CA3 subfields, but not in the cortex (Supplementary Fig. 9g, h). Conditional deletion of REST in excitatory neurons also resulted in neuronal loss in the hippocampal CA1 and CA3 subfields, but not in the cortex (Supplementary Fig. 9i, j). These results suggest that loss of REST promotes neurodegeneration associated with Aβ deposition.

Homozygous REST deletion in a wild-type background (REST cKO mice) resulted in significant neuronal loss in the hippocampal CA1 subfield, but at a lower level than that observed in 3xTg;cKO mice (Supplementary Fig. 9e, f). In contrast, partial REST deletion in a wild-type background (WT;cHET mice) did not result in neuronal loss, whereas 3xTg;cHET mice exhibited significant neuronal loss (Supplementary Fig. 9e, f). These results suggest that AD-type pathology

renders neurons more vulnerable to partial loss of REST. Thus, REST protects against neurodegeneration and preserves the viability of neurons subject to pathogenic Aβ and tau accumulation.

## REST protects against cognitive decline

To determine whether loss of REST might mediate the progression to memory loss and AD, we performed behavioral testing in the 3xTg and J20 mouse models with varying levels of REST expression. 3xTg mice with a conditional heterozygous deletion of REST limited to hippocampal CA1 neurons showed significant impairment in learning and memory in the Morris water maze relative to 3xTg and WT mice (Fig. 8c, d). REST inactivation did not affect general activity in the open field test or swim speed in the Morris water maze; in addition, the ability to locate a visible platform was unaffected (Supplementary Fig. 10a–c). Thus, partial loss of REST function is sufficient to impair learning and spatial memory in 3xTg mice.

To confirm the effects of the conditional *REST* knockout, we assessed aged *3xTg;GT* mice carrying a *REST^{GT}* allele. Partial *REST*

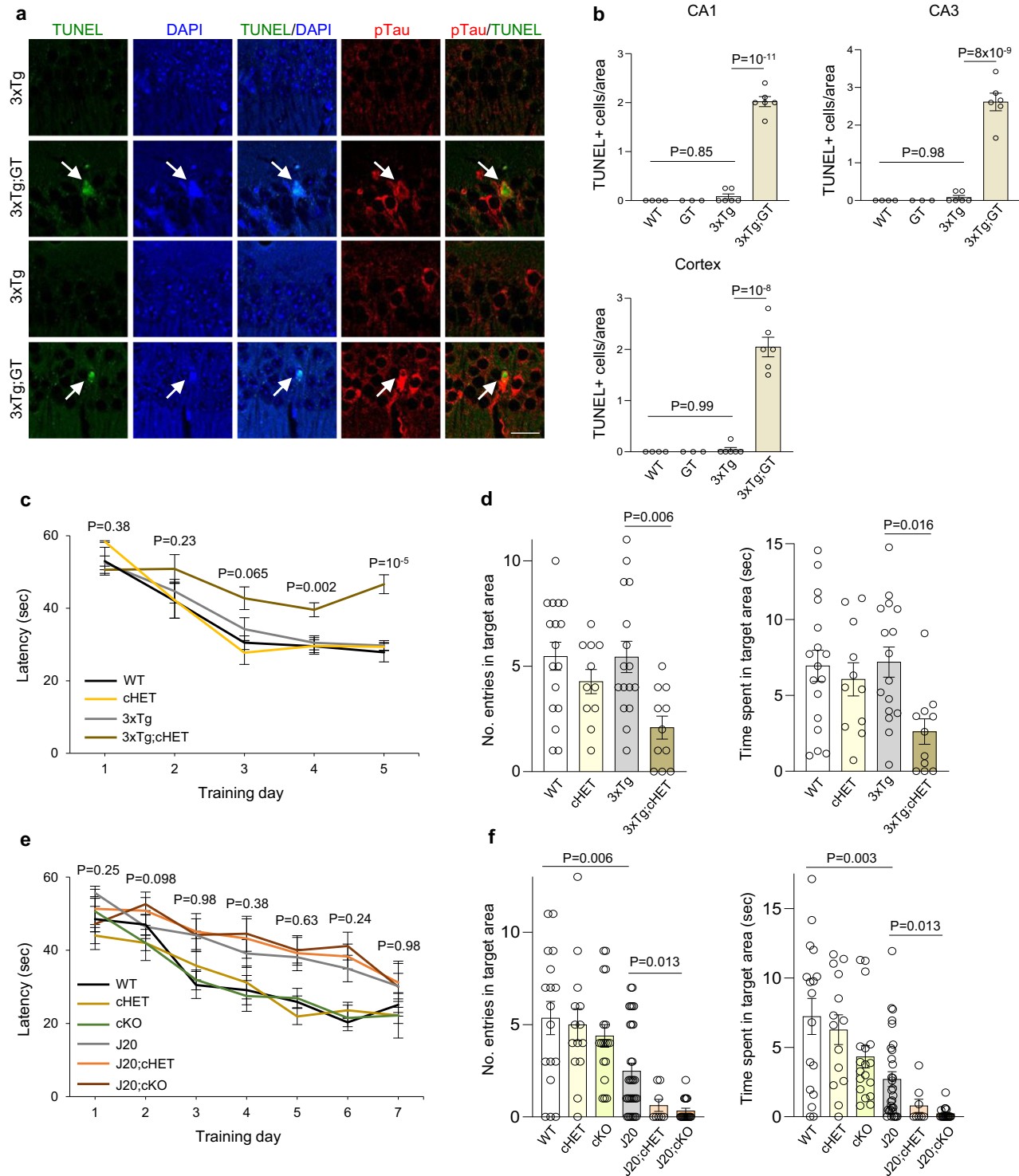

inactivation led to impaired learning and memory retrieval in the Morris water maze (Supplementary Fig. 10d, e). Swim speed and the ability to reach a visible platform were not significantly different in *3xTg;GT* versus *3xTg* mice (Supplementary Fig. 10f, g). In the novel object recognition test of memory, aged 3xTg mice spent more time exploring a novel object than an object they were already familiar with, whereas aged *3xTg;GT* mice showed significanty reduced novel object recognition (Supplementary Fig. 10h). Thus, partial *REST* inactivation in 3xTg mice impairs learning and memory.

The effects of REST on cognitive function were also assessed in J20 mice with robust Aβ deposition. J20 mice exhibited impaired learning and spatial memory relative to WT mice at 12–14 months of age, as determined in the Morris water maze (Fig. 8e, f). Conditional deletion of REST in *J20;cKO* mice did not further impair learning (Fig. 8e) but resulted in significantly greater impairment in memory retrieval (Fig. 8f). Both *J20;cKO* and *J20* mice showed similar behavior in the open field test, as well as similar swim speed and ability to locate a visible platform (Supplementary Fig. 10i–k). Taken together, these results suggest that REST protects against memory loss associated with Aβ deposition and tau accumulation in mouse models.

We next asked whether REST expression levels predict cognitive function in aging humans. To address this question, nuclear REST levels were determined in prefrontal cortical pyramidal neurons by immunofluorescence microscopy in subjects who were enrolled in the

**Fig. 8 | Loss of REST accelerates neurodegeneration and cognitive decline in AD mice. a, b** Increased neurodegeneration in REST-deficient 3xTg mice. **a** TUNEL labeling (green), immunolabeling for p-Ser202 tau (pTau) (antibody CP13, red), and DAPI labeling for DNA identifies TUNEL-positive nuclei in neurons with NFT-like tau pathology in hippocampal CA1 (top 2 panels) and CA3 (lower 2 panels) sectors in 29-month-old 3xTg;GT but not 3xTg mice. **b** Quantification of TUNEL-positive cells in 27–29-month-old WT ($n = 4$), GT ($n = 3$), 3xTg ($n = 6$) and 3xTg;GT ($n = 6$) mice. P-values were generated by two-way ANOVA with Tukey's post-hoc test. Scale bar, 25 μm. **c** Time course of spatial learning in the Morris water maze for 17–18-month-old mice of the indicated genotypes. **d** Memory retrieval in the probe trial of the Morris water maze. Shown are the numbers of entries in the target (platform) area and time spent in target (platform) area for 17–18-month-old mice of the indicated genotypes. Two-way ANOVA revealed no significant interaction between REST genotype (+/+, cHET) and AD pathology (WT, 3xTg): $F_{(1, 51)} = 2.41$, $P = 0.12$. One-

way ANOVA with Tukey's post-hoc test was then conducted and the P-values are shown. Panels **c, d**: $n = 17$ WT, $n = 11$ REST cHET, $n = 16$ 3xTg, and $n = 11$ 3xTg;cHET mice. **e** Time course of spatial learning in the Morris water maze for 12–14-month-old mice of the indicated genotypes. REST loss-of-function alleles do not further impair learning in J-20 mice. **f** Memory retrieval in the probe trial for 12–14-month-old mice of the indicated genotypes. J-20 mice have impaired memory retrieval relative to WT mice by two-tailed Mann–Whitney U test. Two-way ANOVA revealed no significant interaction between REST genotype (+/+, cHET, cKO) and AD pathology (WT, J20): $F_{(2,100)} = 0.82$, $P = 0.44$. Kruskal–Wallis one-way ANOVA with Dunn's post-hoc test was then conducted and the $P$-values are shown. Panels **e, f**: $n = 17$ WT, $n = 14$ REST cHET, $n = 18$ REST cKO, $n = 31$ J20, $n = 8$ J20;cHET and $n = 18$ J20;cKO mice. **b, d, f** Shown are individual values and the mean ± S.E.M. **c, e** Shown are the mean ± S.E.M, and $P$-values for planned comparisons by the two-tailed unpaired $t$-test. Source data are provided as a Source Data file.

ROSMAP longitudinal study of aging and AD. Mutivariate regression analysis was performed to determine relationships between demographic characteristics (age, gender, years of education), apoE genotype, pathological changes (amyloid, tangles, Lewy bodies, macroinfarcts and microinfarcts), REST levels, and an index of global cognitive function in 70–102 year-old individuals with varying cognitive function. Neurofibrillary tangle density was the most significant negative predictor of global cognitive function proximal to death. In contrast, the mean REST level in prefrontal cortical neurons was the most significant positive predictor of global cognitive function (Supplementary Data 8). Thus, REST expression is a strong predictor of cognitive preservation during aging.

## AAV-mediated REST overexpression suppresses AD-type pathology

A central question is whether elevating REST expression in mice with AD-type pathology can suppress disease progression. We employed an adeno-associated virus 9 (AAV9) vector to transduce the human *REST* cDNA under the control of the neuronal promoter CamKII in the hippocampus of 11-month-old 3xTg mice using stereotactic intracranial injection. Ten weeks following injection, REST mRNA expression was elevated by approximatively 2.5-fold in the hippocampus of 3xTg mice that received AAV9-REST compared to AAV9-CTRL (Supplementary Fig. 11a), resulting in increased REST protein expression (Supplementary Fig. 11b, c). REST overexpression resulted in a striking suppression of tau pathology in 3xTg mice as determined by immunolabeling for pSer202 tau (antibody CP13) and misfolded tau (antibody MC1) (Fig. 9a–d). This occurred despite a modest increase in transgene expression (Supplementary Fig. 11f). Furthermore, REST overexpression rescued the advanced tau pathology in 3xTg;GT mice with a partial REST deletion (Fig. 9e, f). REST overexpression also rescued the elevated *Cdk5*, *Gsk3β*, and *Daxx* mRNA levels in 3xTg;GT mice (Fig. 9g). Thus, modestly elevated expression of REST in the 3xTg hippocampus suppresses tau pathology and rescues the phenotype of the *REST* genetrap allele.

We next asked whether REST overexpression affects Aβ deposition in the J20 mouse model. J20 mice injected with AAV9-REST showed an approximately 1.7-fold elevation of REST mRNA expression in the hippocampus, resulting in increased REST protein expression (Supplementary Fig. 11a, d, e). There was no significant change in human APP transgene expression (Supplementary Fig. 11f). REST overexpression had a highly significant effect on Aβ deposition in the hippocampus, reducing Aβ plaque burden by 63% (Fig. 9h, i). These results suggest that modest REST overexpression leads to robust suppression of tau and Aβ pathology in AD mouse models.

## Discussion

These observations identify a neurodegeneration checkpoint that is activated by the onset of Aβ and tau accumulation in the aging brain. This integrated cellular response is mediated by the transcriptional

repressor REST, which is activated in cognitively-intact aging individuals upon accumulation of misfolded proteins, such as Aβ and tau. Conversely, the REST-mediated response is lost in association with cognitive decline and AD. In AD mouse models, genetic deletion of *REST* accelerates Aβ and tau pathology, neurodegeneration and memory loss, whereas modest REST overexpression with an AAV vector strongly suppresses AD-type pathology. Hence, progression to AD may require pathological changes together with an impaired neurodegeneration checkpoint response. The neurodegeneration checkpoint provides a potential mechanistic framework for understanding the prolonged prodromal period that can occur from the onset of Aβ deposition to the appearance of clinical symptoms and ultimately AD[31–33].

Statistical modeling of longitudinal cognitive trajectories in human subjects suggests that the degree of REST activation in the brain correlates with cognitive preservation during aging (Supplementary Data 8). We find that in both AD mouse models, genetic inactivation of REST accelerates cognitive decline. Notably, loss of a single *REST* allele in 3xTg and J20 mice was sufficient to accelerate memory loss, a change that is comparable to the overall level of REST reduction in affected regions of the AD brain. Thus, the degree of REST activation may be an important determinant of cognitive preservation and disease progression during aging.

## Genes that mediate the neurodegeneration checkpoint response

Our findings suggest that there is an early window of REST activation in AD mouse models during the intial stages of Aβ and tau accumulation. REST levels decline in more advanced stages of pathology characterized by substantial plaque burden and neurofibrillary tau accumulation. The dramatic phenotype of REST-deficient mice suggests that REST loss-of-function may contribute to the progression of both Aβ deposition and phospho-tau accumulation. REST directly targets and represses genes that mediate phospho-tau accumulation in 3xTg mice, including the tau kinases GSK3β and CDK5 which have been implicated in neurofibrillary tangle formation. In addition, REST represses the expression of multiple components of the γ-secretase enzyme complex that generates Aβ. In contrast to the tau kinases, it is unclear whether inhibition of γ-secretase components reflects a direct or indirect effect of REST on these genes. Interestingly, REST did not show significant binding to these genes on ChIP-seq. It is possible, however, that long-distance cis or trans enhancer interactions, or effects on other transcription factors or microRNAs, might contribute to the repression of γ-secretase genes. Thus, REST is likely to suppress the progression of amyloid and tau-related pathology through multiple mechanisms of action.

The mechanisms that link amyloid and tau pathology in the pathogenesis of AD have been elusive. Interestingly, REST inactivation led to the accumulation of potentially pathogenic phospho-tau epitopes in J20 APP-transgenic mice, a mouse model that typically shows

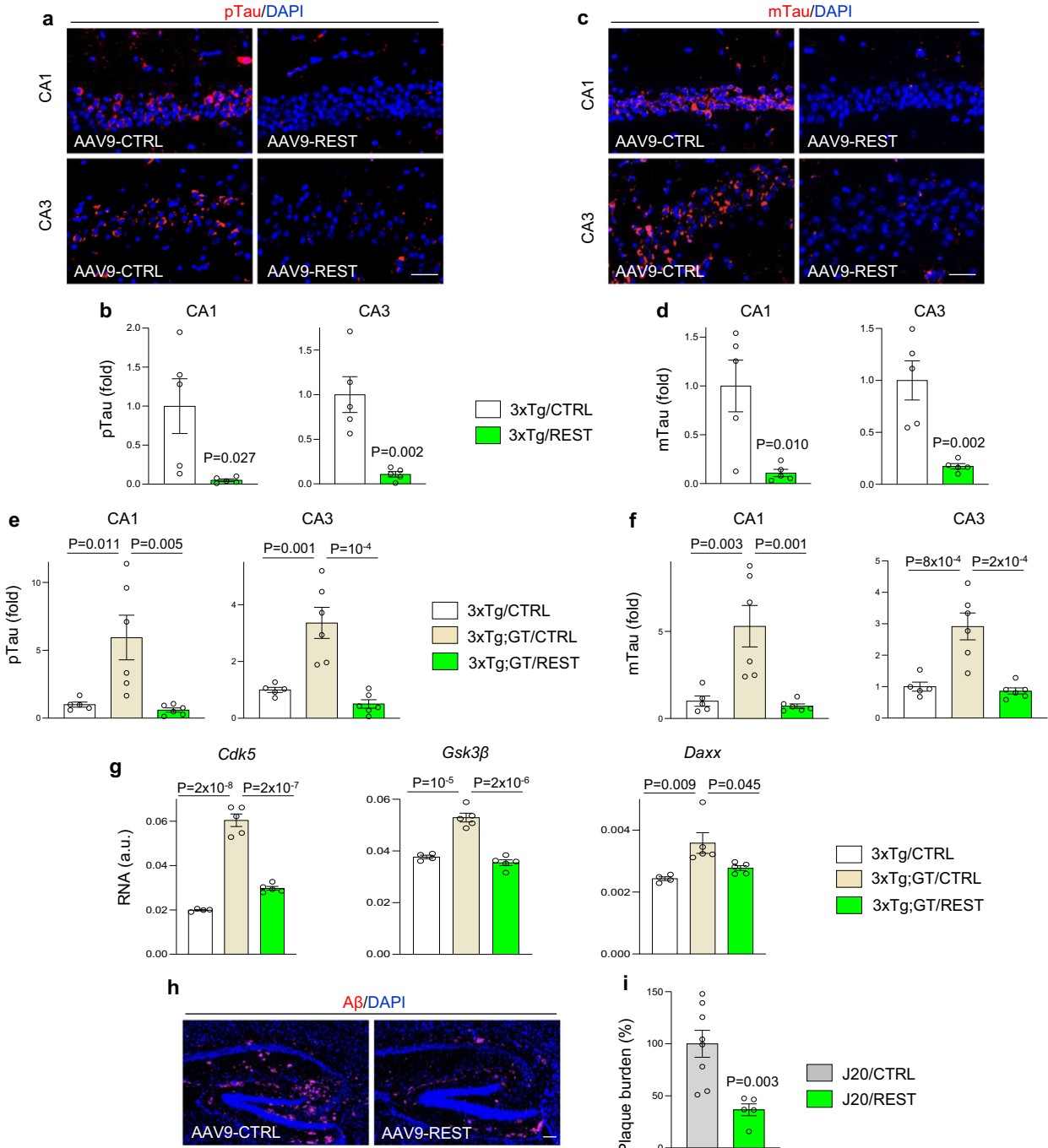

**Fig. 9 | Human REST overexpression suppresses AD-type pathology in mice.**
**a**, **c** Immunofluorescent labeling of the early tau pathology marker pSer202-tau (CP13 antibody, red; pTau; **a**) or misfolded tau (MC1 antibody, red; mTau; **c**) and DNA (blue) in the hippocampal regions CA1 and CA3 of 3xTg mice that had received an intracranial delivery of AAV9-REST or AAV9-CTRL at the age of 11 months and were sacrificed 10 weeks later. Overexpression of human REST leads to a marked suppression of tau pathology. **b**, **d** Quantification of pSer202-tau (pTau; **b**) or misfolded tau (mTau; **d**) levels in CA1 and CA3 regions of the hippocampus. $n = 5$ 3xTg/AAV9-CTRL, $n = 6$ 3xTg/AAV9-REST mice. **e**, **f** REST overexpression rscues the advanced tau pathology in 3xTg;GT mice. Quantification of pSer202-tau (pTau; **e**) or misfolded tau (mTau; **f**) levels in CA1 and CA3 regions of the hippocampus in $n = 5$ 3xTg/AAV9-CTRL, $n = 6$ 3xTg;GT/AAV9-CTRL and $n = 6$ 3xTg;GT/AAV9-REST mice that had received an intracranial delivery of AAV9-REST or AAV9-CTRL at the age of 11 months and were

sacrificed 10 weeks later. **g** qRT-PCR analysis of *Cdk5*, *Gsk3β* and *Daxx* RNA levels in the hippocampus of 3xTg mice that received intracranial injections of AAV9-CTRL ($n = 4$) and 3xTg;GT mice that received injections of AAV9-CTRL ($n = 5$) or AAV9-REST ($n = 5$) at the age of 11 months and were sacrificed 10 weeks after injection. **h** Immunofluorescent labeling of Aβ (red) and DNA (blue) in the hippocampus of J20 mice that had received an intracranial delivery of AAV9-REST or AAV9-CTRL at the age of 14 months and were sacrificed 10 weeks later. Overexpression of human REST leads to a marked suppression of Aβ pathology. **i** Quantification of Aβ plaque burden in the hippocampus, $n = 8$ J20/AAV9-CTRL and $n = 5$ J20/AAV9-REST mice. The data is shown as percentage change in Aβ plaque burden relative to J20/AAV9-CTRL mice. **a**, **c**, **e**, **f** Individual values as well as the mean ± S.E.M are shown. *P*-values for two-tailed unpaired *t*-tests (**b**, **d**, **i**) or one-way ANOVA with Tukey's post-hoc tests (**e**, **f**, **g**) are shown. Scale bars, 25 μm. Source data are provided as a Source Data file.

robust amyloid deposition but minimal phospho-tau accumulation. Hence, the REST-mediated checkpoint response may prevent Aβ from giving rise to phospho-tau accumulation, potentially contributing to the prolonged prodromal period of Aβ accumulation in the brain prior to the spread of neurofibrillary tangles and memory loss.

Increased REST binding to genes that regulate cell death pathways was also observed in the setting of early AD-type pathology in 3xTg mice. These include the inflammation-associated proapoptotic gene *Daxx*, the transcription factor *Foxo3*, and *Casp9*, a mediator of the apoptotic cascade. REST has also been shown to target cell death-related genes in the aging human brain[13]. These findings are consistent with the protective effect of REST against neuronal cell death in both of the AD mouse models examined in this study.

REST induction by early AD-type pathology also results in the targeting of multiple cell cycle genes, including *Cdk5*[34]. This could potentially suppress aberrant cell cycle initiation in postmitotic neurons, a pathogenic mechanism that has been implicated in AD[34,35]. Our findings also implicate REST in the regulation of metabolic genes, as well as genes involved in synaptic structure and signaling, raising the possibility that REST may suppress hypermetabolism and hyperexcitation, both of which have been associated with cognitive impairment[36]. In addition, REST has been shown to suppress some immune genes in the hippocampus[37]. Hence, REST activation may suppress multiple pathogenic pathways to preserve neuronal viability and function during aging.

### REST activation and the unfolded protein response

The neurodegeneration checkpoint appears to be induced by proteostatic changes associated with early AD pathology. In particular, activation of the unfolded protein response potentiates β-catenin signaling leading to robust induction of nuclear REST. REST is a transcriptional target of β-catenin[13,38], which has been implicated in the induction of REST in the aging human brain[13]. Concurrent activation of the UPR, β-catenin and REST was detected in human cortical neurons in association with early AD pathology, as well as in AD mouse models. Our findings also implicate a role for the PERK kinase that participates in the initiation of the UPR. Activation of PERK has been associated with diffuse accumulation of phosphorylated tau prior to neurofibrillary tangle formation in AD[20]. Further insight into the regulation of the UPR-β-catenin-REST pathway in aging neurons will be important for understanding the loss of this neuroprotective pathway in individuals who progress to AD.

It will be of interest to determine whether misfolding of other pathogenic proteins associated with age-related neurodegeneration, such as α-synuclein and TDP-43, might also activate the REST-mediated neurodegeneration checkpoint. Early loss of proteostasis in subpopulations of affected neurons is a defining feature of many neurodegenerative disorders that precedes the onset of clinical disease. A greater understanding of this checkpoint pathway might lead to new therapeutic approaches based on the physiology of neuroprotection in the aging brain.

## Methods

Postmortem human brain material was procured in accordance with institutional guidelines and was approved by the Harvard Medical School Institutional Review Board. Animal housing and experimental procedures were approved by the Institutional Animal Care and Use Committee of Harvard Medical School.

### Brain sample procurement and description

Tissue samples were procured from the Rush Alzheimer's Disease Center (RADC). Tissue samples (both paraffin-embedded and frozen) from the RADC were derived from participants in the Religious Orders Study (ROS) or Rush Memory and Aging Project (MAP) (together referred to as ROSMAP) at the RADC; these are longitudinal, clinical–pathologic studies of ageing, cognitive decline and AD[39]. Study participants agreed to comprehensive annual clinical and neuropsychological evaluation and to brain donation at death. Informed consent, an Anatomic Gift Act, and a repository consent were obtained and the study was approved by an Institutional Review Board (IRB) of Rush University Medical Centre. ROSMAP data can be requested at https://www.radc.rush.edu.

To assess cognitive function, 21 cognitive function tests were used, 11 of which directly informed on clinical diagnoses of Alzheimer's dementia, mild cognitive impairment (MCI) and no cognitive impairment (NCI) as previously described[40,41]. A global measure of cognition was computed from 19 independent test scores (7 assessments of episodic memory, 3 measures of semantic memory, 3 measures of working memory, 2 measures of perceptual orientation, and 4 measures of perceptual speed; see https://www.radc.rush.edu/docs/var/detail.htm?category=Cognition&subcategory=Global+cognition&variable=cogn_global)). APOE genotype was determined as previously reported[42]. The follow-up rate exceeds 95% and the autopsy rate exceeds 90%.

All individuals who underwent autopsy were subject to a uniform structured neuropathologic evaluation of AD, including assessment of global pathology, a quantitative summary of AD pathology derived from standardized counts of neuritic and diffuse plaques, and neurofibrillary tangles, determined by microscopic examination of silver-stained slides from 5 regions, Neuritic plaque pathology (CERAD score), the density and distribution of neurofibrillary tangles (Braak stage), and a composite measure of neurofibrillary tangles and neuritic plaques (NIA/Reagan score) (https://www.radc.rush.edu/docs/var/detail.htm;jsessionid=329C70639AA50EAE971DBEC631D06872?category=Pathology&subcategory=Alzheimer%27s+disease&variable=gpath). Measures of neuropathology at AD pathology also included β-amyloid load and PHFtau tangle density from immunostains of 8 brain regions. In adition, we examined Lewy bodies, and neocortical, macroinfarcts and microinfarcts as previously reported[43].

Using a sample of $n = 518$ NCI cases and $n = 501$ AD dementia cases from the ROSMAP cohort, we stratified the cases based on clinical diagnosis (NCI or AD) and the stage of AD pathology, as reflected by the global AD pathological burden (*gpath*): no AD pathology (*gpath* values below the 15th percentile: $0 \leq gpath \leq 0.09$), early AD pathology (*gpath* values between the 15th and the 40th percentile: $0.09 < gpath \leq 0.49$), mid AD pathology (*gpath* values between the 40th and the 75th percentile: $0.49 < gpath \leq 1.2$) and late AD pathology (*gpath* values above the 75th percentile: $gpath > 1.2$). The differences between the 4 stages of pathology were statistically significant for the measure of global pathology (gpath), as well as CERAD, Braak and NIA/Reagan scores (Supplementary Fig. 1a). The distribution of each level of AD pathology among NCI and AD cases is shown in Supplementary Fig. 1b.

Postmortem human samples ($n = 82$ NCI and $n = 63$ AD cases) were randomly selected from a larger pool of ROSMAP samples, based on pre-set criteria that were equally applied to all samples: age (70–102), low postmortem intervals (typically 6–10 h or less) and the availability of paraffin-embedded tissue. The randomly-selected postmortem samples were then allocated to the experimental groups based on the cognitive status of each case -- no cognitive impairment (cogdx=1; group 1) vs. Alzheimer's disease (cogdx=4; group 2). Within each of these 2 groups, the samples were further sub-divided into 4 groups (no pathology, early pathology, mid pathology and late pathology) using the following cutoffs for the global AD pathological burden (gpath) variable: no AD pathology ($0 \leq gpath \leq 0.09$), early AD pathology ($0.09 < gpath \leq 0.49$), mid AD pathology ($0.49 < gpath \leq 1.2$) and late AD pathology (gpath > 1.2).

Nuclear REST levels in PFC neurons were assessed in $n = 82$ NCI and $n = 63$ AD cases from ROSMAP (Fig. 1a–c), partitioned according to clinical diagnosis and AD pathology into 8 groups (see above). The

mean age at death (in years) was similar across groups: 84 ± 7 (NCI, no path), 87 ± 5 (NCI, early path), 88 ± 7 (NCI, mid path), 87 ± 2 (NCI, late path), 83 ± 5 (AD, no path), 82 ± 5 (AD, early path), 85 ± 3 (AD, mid path), and 87 ± 5 (AD, late path). The mean post-mortem interval (PMI, in hours) was similar across groups: 7 ± 3 (NCI, no path), 7 ± 5 (NCI, early path), 6 ± 1 (NCI, mid path), 6 ± 2 (NCI, late path), 10 ± 9 (AD, no path), 5 ± 4 (AD, early path), 5 ± 4 (AD, mid path), 6 ± 3 (AD, late path). The aging cohort was comprised of $n = 56$ males (38.6%) and $n = 89$ females (61.3%): NCI cases with no pathology ($n = 21$, 10M, 11F), early pathology ($n = 30$, 10M, 20F), mid pathology ($n = 26$, 11M, 15F) and late pathology ($n = 5$, 2M, 3F), as well as in AD cases with no pathology ($n = 3$, 1M, 2F), early pathology ($n = 3$, 2M, 1F), mid pathology ($n = 13$, 6M, 7F) and late pathology ($n = 44$, 14M, 30F). Two-way ANOVA analyses showed that gender did not affect REST expression in the aging population, nor did it interact with pathology or diagnosis to predict REST expression (see Supplementary Data 1).

### Immunofluorescence analysis of human brain

For immunofluorescence analysis of the prefrontal cortex (Brodmann areas 9, 10 and 47), we first deparaffinized the paraffin-embedded post-mortem human brain samples (by immersion in 2 xylene baths for a total of 10 min, followed by a 5-min immersion in a 50% Xylene:50% ethanol solution). The sections were then rehydrated by immersion in solutions of decreasing concentrations of ethanol in water (95%, 90%, 70%, and 50%) and then placed in water. Sections then underwent antigen retrieval using the Diva decloaker (BioCare, USA). Sections were blocked with 2% BSA, 2% fetal bovine serum (FBS) and 0.1% Triton X-100 in PBS for 45 min at room temperature. Primary antibodies were diluted in 2% BSA, 2% FBS and 0.1% Triton in PBS. Following overnight incubation at 4 °C, sections were washed three times with PBS. Secondary antibodies, diluted in 2% BSA, 2% FBS and 0.1% Triton in PBS were either biotin-coupled (1:200, Vector Labs, USA) or coupled to Alexa fluorophores (1:300, Invitrogen). Sections were incubated with 1% Sudan Black in 80% ethanol for 15 min at room temperature to suppress lipofuscin autofluorescence. Sections were then mounted with Pro-Long anti-fade mounting medium with DAPI (Invitrogen) and then imaged using confocal microscopy.

The specificity of the anti-REST antibody (Bethyl, IHC-00141) was confirmed by us previously[13]. Briefly, labeling of human neural cells in which *REST* was knocked down or overexpressed led to a significant loss, and a significant increase, in REST nuclear immunolabeling, respectively. The qualitative and quantitative pattern of REST immunolabeling for the REST Bethyl IHC-00141 antibody was very similar to those obtained with 2 independent REST antibodies by immunofluorescence[13]. When the REST Bethyl IHC-00141 antibody is pre-incubated with excess of the REST immunizing (blocking) peptide (corresponding to aa 1000-1097 of REST), REST immunolabeling is largely abrogated (ref. [13] and see Supplementary Fig. 1c). No signal was detected when the REST Bethyl IHC-00141 primary antibody was omitted from the staining protocol.

To assess the levels of nuclear REST in neurons of the prefrontal cortex, we performed confocal immunofluorescence with triple immunolabelling for REST (green), the neuronal marker MAP2 (magenta), and DAPI labeling of nuclei (blue). Pictures were acquired in 4–6 randomly selected fields of the prefrontal cortex, using an Olympus FluoView™ LV1000 confocal microscope and a 40x objective, UPLAPO40xOI2. We then quantified the nuclear REST fluorescence intensity in 25–125 cortical neurons– identified by MAP2 co-labeling – and derived the mean nuclear REST fluorescence intensity for each case. The values of nuclear REST fluorescence intensity were corrected by subtracting the average background intensity for REST in each field (measured outside cells). Nuclear REST levels in cortical neurons were assessed in $n = 82$ NCI and $n = 63$ AD cases (Fig. 1b). The investigators were blind to sample diagnosis.

### Multivariate regression analysis of global cognitive function

To determine whether REST, age, gender, education, the inheritance of an apoE4 allele, and the presence of neuropathology predict cognitive function in the aging population, we conducted a multivariate regression analysis to determine which independent variables (REST, age, gender, education, apoE4 genotype, neuropathological measures) predict global cognitive function (dependent variable). Measures of neuropathology, nuclear REST levels in post-mortem prefrontal cortical neurons, and the age when the cognitive tests were conducted were obtained for each individual case. ApoE4 genotype was coded as binary variable (1- carrier of at least one apoE4 allele; 0- no apoE4 allele). We analyzed $n = 145$ cases for which REST nuclear levels in PFC pyramidal neurons were measured ($n = 82$ NCI and $n = 63$ AD cases) and the global cognitive, demographic and neuropathologic scores were available. The age range for these cases is 70–102. REST nuclear levels in cortical pyramidal neurons were standardized (the transformed nuclear REST levels had an average of zero, and a standard deviation of 1). Global cognitive function measurement used is the measurement closest to death for each individual (mean = 1.2, median = 0.8, max = 7.9 years before death). Six different regression analyses were conducted, in which different variables were included as predictors of global cognitive function: (a) nuclear REST levels in cortical pyramidal neurons; (b) age and REST cortical levels; (c) age, AD pathology, and REST levels in cortical neurons; (d) age, AD pathology, infarcts, and REST cortical levels; (e) age, gender, apoE4 genotype, education, AD pathology, and REST cortical levels; or (f) age, gender, apoE4 genotype, education, AD pathology, infarcts, and REST cortical levels (Supplementary Data 8). REST nuclear levels in cortical pyramidal neurons were a significant predictor in all 6 regression models (Supplementary Data 8). Similar conclusions were reached when limiting analyses to individuals that have their final measurement of global cognitive function ≤2 years from death. Multiple linear regression models were fit in GraphPad Prism (version 9.3.1).

### Mice

All mice were housed socially (2–4 animals/cage) in a room with a 12-h light/dark cycle (lights on at 6:00 am), controlled for temperature (18–22 °C) and humidity (40–60%). Food and water were provided ad libitum. Sentinel mice housed in each rack were tested quarterly and confirmed free of pathogens.

Mice carrying floxed alleles of REST flanking exon 2 were obtained from William H. Klein (The University of Texas MD Anderson Cancer Center, Houston, TX, USA) and were described previously[24]. These mice were crossed to either CA1-specific CamKIIα-Cre[22] or forebrain-specific CamKIIα-Cre mice[23] (both obtained from Jie Shen, Brigham and Women's Hospital, Harvard Medical School) to achieve REST conditional inactivation in hippocampus CA1 pyramidal neurons, or cortical and hippocampal excitatory glutamatergic neurons, respectively. As a second strategy for deleting *REST*, we employed a *REST* genetrap (*REST^GT*) allele, in which a β-geo cassette is inserted in the *REST* intron between non-coding exon 1a-c and the first coding exon, exon 2, leading to a null *REST* allele[25]. *REST^{GT/GT}* mice die from embryonic day 9.5–11.5, but *REST^{GT/+}* are viable[25] and have been maintained as long as 29 months. The *REST^GT* line was obtained from Wolfgang Wurst (Helmholtz Center Munich, Germany). The CamKIIα-Cre transgenes are in the C57BL/6J background, and the REST^{fx/fx} alleles is in a hybrid C57BL/6J and 129 Sv/Ev background. The *REST^GT* allele was in a C57BL/6J background. The J20 mice express transgenic mice express a mutant form of the human amyloid protein precursor bearing both the Swedish (K670N/M671L) and the Indiana (V717F) mutations (APPSwInd) in a C57BL/6J background[15]. The 3xTg mice carry *APP^{Swe}* and *tau^{P301L}* mutant transgenes, as well as a *PS1* knock-in mutation[14] and were in a hybrid C57BL/6J and 129Sv/Ev background. The J20 and 3xTg lines were obtained from The Jackson Laboratory.

To generate mutant mice, we typically set up 10–20 females (all littermates derived from the same cross) with 8–12 males (all littermates derived from the same cross). CamKIIα-Cre:REST^fx/fx conditional knockout mice, or controls (REST^fx/fx, REST^fx/+ or CamKIIα-Cre) in either WT, J20 or 3xTg backgrounds (hybrid C57BL/6J and 129Sv/Ev background), as well as REST^GT/+ or REST^+/+mice, in either WT, J20 or 3xTg backgrounds (hybrid C57BL/6J and 129Sv/Ev background) were born at expected Mendelian ratios, were viable and fertile, and did not display any visible alterations. Mice were identified by numbered ear tags and were randomly selected for behavioral studies and histological analyses.

To determine whether REST is involved in the onset and progression of AD pathology and cognitive decline, we examined AD mice (3xTg and J20 models) carrying REST knockout or WT alleles. To determine whether genetic deletion of REST enhances AD pathology, we employed animals with a mild AD phenotype, which thus allow the detection of any enhancement of pathology and cognitive function upon REST deletion. Hemizygous 3xTg mice display a mild accumulation of amyloid and tau[14], and were thus ideal for genetic enhancement studies. These mice were crossed to mice carrying Cre and floxed REST alleles, or mice carrying the REST gene trap allele.

Once the desired genotypes were obtained, mice were subjected to the aging process. Mice were analyzed at defined periods during the aging process. For crosses involving 3xTg, mice were analyzed at 17–18 or 28 months of age, given the milder phenotype of the 3xTg hemizygous mice. Mice with a complete REST deletion in forebrain excitatory neurons (3xTg carrying floxed REST alleles and CamKIIα-Cre) were analyzed at 17–18 months. Mice with a partial (heterozygous) REST deletion (3xTg carrying the REST gene trap allele) were aged to 27–29 months ($27.7 \pm 0.9$ months for 3xTg and $27.5 \pm 0.7$ months for 3xTg;GT).

The J20 mice display a more robust accumulation of amyloid, with onset of plaque formation around 6 months of age. The J20 mice with a complete conditional REST deletion in neurons (J20;cKO) were analyzed at 12–14 months of age ($13.1 \pm 0.7$ for J20; $13.2 \pm 0.8$ for J20;cHET and $13.2 \pm 0.6$ for J20;cKO) whereas J20 mice with a partial, heterozygous REST gene trap deletion (J20;GT) were analyzed at 27–29 months of age ($28.1 \pm 1$ months for J-20 and $27.9 \pm 1$ months for J-20;GT mice).

Most experiments included both male and female mice (information about the gender of all animals used in the various experimental groups is available in the Supplementary Data 1). Unless otherwise stated, gender had no significant effect on the measured dependent variables, nor did it significantly interact with genotype to influence the measured variables (see Supplementary Data 1 for two-way ANOVA analyses of gender and genotype effects, as well as gender x genotype interactions).

## Mouse DNA extraction and genotyping

We collected -0.5–1 cm mouse tails in clean Eppendorf tubes. 500 µL of tail lysis buffer (10 mM Tris pH 8, 100 mM NaCl, 10 mM EDTA, 0.5% SDS) containing 0.4 mg/mL Proteinase K was added to each tube, and the tubes were incubated overnight in a 56 °C water bath. The next day, 500 µL of isopropanol was added to precipitate the DNA, and the tubes were shaken vigorously for 20 s. Tubes were centrifuged for 10 min at $18,000 \times g$, and the isopropanol was carefully removed, while avoiding the DNA pellet. We then added 70% ethanol, and shook the tubes, to wash the DNA pellet. We next centrifuged the tubes for 10 min at $18,000 \times g$. We removed the ethanol and air-dried the DNA pellet for 2–16 h. DNA was re-suspended in 100 µL acetate-EDTA (AE) buffer and placed in a 56 °C water bath overnight. To amplify DNA regions by PCR, we mixed 3 µL of DNA sample with corresponding amounts of forward and reverse PCR primers, PCR master mix, and nuclease-free water, and run the reactions in a thermocycler. Sample loading dye was added to the PCR products, and the samples were run on 1–3% agarose gels

(prepared by dissolving agarose in 1x TAE buffer, to which Gel Red™ was added to allow DNA visualization). We also loaded a 100 bp DNA ladder. Gels were visualized using a UV transilluminator. The PCR primer sequences are shown in Supplementary Data 6.

## Open field

Mice were placed in an open field box (75 cm × 75 cm) and movements were tracked in real-time using the TopScan Lite software (CleverSys Inc.) coupled to a camera. Each mouse was recorded for 10 min, and the mouse average speed and distance traveled were automatically recorded. Mice had no prior exposure to the open field arena (spontaneous test).

## Morris water maze

To assess spatial learning and memory, we trained and tested mice in a large circular pool (1.1 meters in diameter) filled with 21 °C water, which was rendered opaque by the addition of non-toxic white paint. We placed four distinct visual cues (representing different geometric shapes, patterns, and colors) on each wall, to facilitate spatial orientation and the acquisition of spatial memory. Mice were given 4 training trials/day for 5–7 consecutive days. Each training trail lasted for 1 min. Mice were trained to remember the location of a hidden platform, which was submerged 1 inch below the water surface. The location of the hidden platform (SE) remained the same during the 5–7-day training period. If after a 60 s trial, the animal failed to locate the platform, the animal was placed on the platform and allowed to remain on the platform for 15 s. Mice were trained 4 times a day and entered the pool in a randomized order of rotating entrance points (N, S, E, W, NE, and SW). During each training trial, the latency to find the hidden platform was recorded. 24 h after the last training trial, a probe trial was conducted. The platform was removed, and mice entered the arena from the NW location (opposite from the platform). The number of entries in the target area (representing the area where the platform had been located during the training trials); the total time spent in the target area; as well as the time spent in all quadrants, and the swim speed, were recorded during the 60-s probe trial. We also conducted separate trials in which a visible platform (platform elevated above the water level, on which a small red flag had been placed) was presented. Mice were given several training sessions and the time (latency) to reach the visible platform was recorded. Mouse movements, as well as average speed, distance traveled, latency to reach a quadrant or target area, or number of entries in the target area, were tracked in real-time using the TopScan Lite® software (Clever Sys Inc.) and the different measures were automatically recorded.

## Novel object recognition

Mice were placed in the same open field box with two novel identical objects for 10 min and allowed to freely explore the identical objects. The next day, mice were reintroduced in the open field box, and presented with a novel object, as well as one of the 2 objects they explored the previous day. The mice were allowed to explore the objects for 10 min, and their movements were tracked in real-time using the TopScan Lite® software (Clever Sys Inc.) coupled to a camera. The box and items were cleaned with 70% ethanol in between mice. We automatically recorded the time each mouse spent exploring each object, on both day 1 (2 identical objects) or day 2 (one novel object, one old object), and derived an object preference index (representing the percentage of time the mouse explored the object, relative to the time spent exploring both objects).

## Mouse perfusion and brain dissection

Mice were anesthetized with isoflurane and carbon dioxide and then perfused with cold PBS buffer for 20 min. Brains were rapidly removed, and the 2 hemispheres were separated. One hemisphere was dissected into subregions (frontal cortex, temporal cortex, occipital cortex,

hippocampus, and cerebellum); each subregion was placed in a separate Eppendorf tube and snap-frozen in liquid nitrogen, then stored in a −80 °C freezer. The second hemisphere was placed in 4% paraformaldehyde for 48 h. The fixed brain was then processed for paraffin embedding, using standard procedures. Paraffin-embedded blocks were sectioned and 6-micron sections were mounted on glass slides and used for histological analyses.

## Antibodies

The following antibodies were used to detect the REST protein: (1) a rabbit polyclonal IgG that recognizes a region between residues 1050–1097 (C-terminus) of REST (Bethyl laboratories, IHC-00141). (2) The REST C-terminal antibody used for ChIP-seq was a gift from Gail Mandel[21] (Vollum Institute). This antibody has previously been used for ChIP[37]. (3) For ChIP-qPCR we used a second anti-REST antibody, as well as non-specific IgG control (both from Millipore Sigma, Catalog No. 17-641). (4) For detection of mouse REST by IHC/IF, we employed the rabbit polyclonal antibody REST14[44], a generous gift from Jenny Hsieh (University of Texas at San Antonio).

To detect different tau species, we used the following antibodies: (1) a mouse monoclonal antibody that recognizes phosphorylated Ser 202 tau (CP13 clone; generous gift from Peter Davies, Albert Einstein College of Medicine, NY). (2) a mouse monoclonal antibody raised against a conformationally altered form of tau in AD (MC1 clone; from Peter Davies). (3) a mouse monoclonal antibody raised against phosphorylated Ser 396 tau (PHF1 clone; from Peter Davies). (4) a mouse monoclonal antibody that recognizes phosphorylated Ser202/Thr205 tau (clone AT8; ThermoFisher Scientific, Catalog No. MN1020). (5) a mouse monoclonal antibody that recognizes phosphorylated Thr231 tau (clone AT180; ThermoFisher Scientific, Catalog No. MN1040). (6) a rabbit polyclonal antibody that recognizes phosphorylated Thr217 tau (ThermoFisher Scientific, Catalog No. 44-744). (7) a mouse monoclonal antibody that recognizes all tau species (total tau; clone tau-5; ThermoFisher Scientific, Catalog No. AHB0042).

Additional primary antibodies were as follows: anti-human Aβ rabbit monoclonal IgG antibody (Cell Signaling, Cat. No. 8243); anti-human APP mouse monoclonal IgG antibody (clone 6E10; Covance, Catalog No. SIG-39320); anti-actin mouse monoclonal IgG antibody (clone ACTN05 (C4); ThermoFisher Scientific, Catalog No. MA5-11869); anti-NeuN mouse monoclonal IgG antibody (clone A60, Millipore, MAB377); anti-MAP2 goat polyclonal IgG antibody (PhosphoSolutions, Catalog. No. 1099-MAP2); anti-CDK5 mouse monoclonal IgG antibody (clone 4E4; Novus Bio, Catalog No. NBP2-37602); anti-GSK3β mouse monoclonal IgG antibody (clone D5C5Z; Novus Bio, catalog No. NBP1-47470S); anti-PS1 C-terminal (CTF) rabbit monoclonal IgG antibody (clone EP2000Y; Abcam, Catalog No. ab76083); anti-PS1 N-terminal (NTF) rabbit polyclonal IgG antibody (231-f; made in the Yankner lab); anti-Nicastrin mouse monoclonal IgG antibody (clone 9C3; Biolegend, Catalog No. 852301); anti-Nicastrin rabbit polyclonal IgG antibody (Sigma Millipore, Catalog No. N1660); anti-PEN2 rabbit polyclonal IgG antibody (ProScience, Catalog No.3981); anti-PEN2 rabbit monoclonal IgG antibody (clone EPR9200; Abcam, Catalog No. ab154830); anti-PEN2 rabbit polyclonal IgG (ProScience, Catalog No. 3981); anti-Transferrin receptor mouse monoclonal IgG antibody (clone H68.4; ThermoFisher Scientific, Catalog No. 13-6800); anti-BiP/GRP78 mouse monoclonal IgG (clone C38; ThermoFisher Scientific, clone C38, Catalog No. 14-9768-82); anti-β-catenin goat polyclonal IgG (R&D Systems, Catalog No. AF1329); non-specific rabbit IgG antibody (Sigma Millipore, Catalog No. 17-641); and anti-FLAG mouse monoclonal IgG1 antibody (clone M2; Sigma Millipore, Catalog No. F3165).

The secondary antibodies were as follows: Alexa Fluor 647 donkey anti-rabbit IgG (ThermoFisher Scientific, Catalog No. A31573);Alexa Fluor 488 donkey anti-rabbit IgG (ThermoFisher Scientific, Catalog No. A21206); Alexa Fluor 694 donkey anti-mouse IgG (ThermoFisher Scientific, Catalog No. A21203); Alexa Fluor 647 donkey anti-goat IgG (ThermoFisher Scientific, Catalog No. A21447). See Supplementary Data 7 for details about specific uses and dilutions for each antibody.

## Immunofluorescence staining of the mouse brain

Paraffin-embedded mouse brain blocks were sectioned, and the sections were mounted on glass slides. We deparaffinized the sections by immersion in 2 xylene baths for a total of 10 min, followed by a 5-min immersion in a 50% Xylene-50% ethanol solution. The sections were then rehydrated by immersion in solutions of decreasing concentrations of ethanol (95%, 90%, 70%, and 50%) and then placed in water. Sections then underwent antigen retrieval using the Diva decloaker (BioCare, USA). Sections were blocked with 3% BSA, 3% fetal bovine serum (FBS) and 0.1% Triton X-100 in PBS for 45 min at room temperature. Primary antibodies were diluted in 3% BSA, 3% FBS and 0.1% Triton in PBS. Following overnight incubation at 4 °C, sections were washed three times with PBS. Secondary antibodies, diluted in 3% BSA, 3% FBS and 0.1% Triton in PBS were either biotin-coupled (1:200, Vector Labs, USA) or coupled to Alexa fluorophores (1:300, Invitrogen). After three 10-min washes with PBS, sections were mounted with Pro-Long anti-fade mounting medium with DAPI (Invitrogen) and then imaged using confocal microscopy. For NeuN labeling, we incubated sections with an anti-mouse biotinylated IgG (VectorLabs) for 1 h, followed by 3 washes in PBS (1 min each) and the addition of avidin-streptavidin-HRP-coupled complex (1:200 in 2% BSA and 0.1% Tritonx100 in PBS, VectorLabs). After 3 washes with PBS we added diaminobenzidine (DAB) substrate (prepared by dissolving DAB and peroxide tablets in PBS; Sigma-Aldrich) and incubated for several minutes, until a brown precipitate forms. Sections were then washed with PBS, dehydrated with increasing ethanol concentrations (50%, 70%, 90%, 95%, 100%), followed by incubation with a 50% ethanol-50% xylene solution and two immersions in 100% xylene (5 min each). Sections were mounted with a hydrophobic mounting medium (Permount). Multiple confocal images were acquired using an Olympus Fluoview Confocal Microscope FV1000 in the hippocampus and the cortex of each mouse, spanning both the anterior and posterior aspects. For DAB-stained sections, we acquired pictures using a brightfield microscope coupled with a camera.

## Assessment of Aβ and tau pathology

For analysis of Aβ plaque burden, pictures of Aβ immunoreactivity in the hippocampus or the cortex were processed using a macro developed for use with Fiji/ImageJ. Confocal pictures were all saved in the same folder and were all automatically opened in Fiji and processed serially. For each picture, the background was subtracted (rolling ball radius was set for 25). Pictures then underwent de-noising, using a Gaussian blur filter (radius = 1 pixel). The images were then thresholded using the Default Fiji threshold set at 120. Particles with a minimal size of 5 μm² were retained and their number, average size, mean fluorescence intensity was automatically recorded for each picture in an Excel file. To calculate the Aβ plaque burden, the total area occupied by Aβ plaques was divided by the area of the selection. Three coronal sections (6 micron thick) were sampled for each animal, in the rostral, intermediate and ventral CA1 and the adjacent cortex; the sections were 240 microns apart. Two 20× images were acquired within the CA1 or CA3 sectors, as well as the cortex for each section (six 20× images were acquired each for CA1, CA3 and cortex/animal), using an Olympus FluoView™ LV1000 confocal microscope. The investigators were blind to sample genotypes.

The average Aβ burden was obtained by averaging the Aβ plaque density (area occupied by Aβ plaques divided by the total area analyzed) in all pictures acquired in the hippocampus (or the cortex) for each animal. For analysis of tau pathology, pictures of p-Ser202 tau (CP13) or MC1 tau immunoreactivity in the hippocampus (regions CA1, CA3) were processed using a macro developed for use with Fiji/ImageJ. Confocal pictures were all saved in the same folder and were all

automatically opened in Fiji and processed serially. Pictures underwent de-noising, using a Gaussian blur filter (radius = 1 pixel). The images were then thresholded using the Default Fiji threshold set at 150. The number of tau-positive neurons within selected CA1, CA3 or cortical areas was then manually counted for each thresholded picture, and the area of each region (CA1, CA3, and cortex) was measured. For each picture, we calculated the average density of tau-positive neurons (the total number of tau-positive neurons divided by the area of the region). The average tau-positive neuron densities in the hippocampal CA1 or CA3 sectors, or in the cortex (layers 3–5) was calculated for each animal, by averaging all pictures acquired in each sector of the hippocampus or the cortex. The investigators were blind to sample genotypes.

## Assessment of hippocampal and cortical neuron density

We used hematoxylin and eosin (H&E)-stained brain sections to quantify the density of CA1, CA3 and cortical neurons. Neurons in the hippocampal CA1 and CA3 sectors were easily identifiable by their location and size. In the cortex, we identified neurons in layers 3–5 by their size and morphology. Neurons have larger nuclei compared to astrocytes (smaller nuclei) and microglia (very small and dense nuclei). For quantification of neuronal density, randomly selected areas within the hippocampus or the cortex, and at various depths within these structures, were imaged at ×20 magnification. The number of cells (using either hematoxylin and eosin or NeuN labeling) was determined using the MetaMorph® image analysis software and was then divided by the area occupied by these cells. The investigators were blind to sample genotype.

## TUNEL and p-Ser202-tau double labeling

Paraffin slides first underwent deparaffinization by immersion in 2 xylene baths for a total of 10 min, followed by a 5-min immersion in a 50% Xylene-50% ethanol solution. The sections were then rehydrated by immersion in solutions of decreasing concentrations of ethanol (95%, 90%, 70%, and 50%) and then placed in water. TUNEL labeling was performed using the ApopTag® Fluorescein in Situ Apoptosis Detection Kit (Millipore Sigma, Cat. No. S7110) according to the manufacturer's instructions. Following TUNEL labeling, sections were incubated in 1% PFA for 5 min and then washed three times with PBS. They then underwent pSer202-tau labeling with the CP13 antibody, as detailed above. For analysis of TUNEL and CP13 labeling, multiple fields in the hippocampus (CA1 and CA3) and the cortex (layers 3–5) were acquired using an Olympus Fluoview confocal microscope at 20X magnification and 1.5 optical zoom. The density of TUNEL-positive nuclei was quantified in each picture, and an average density of TUNEL-positive nuclei was calculated for each structure (CA1, CA3, and cortex) and animal. The investigators were blind to sample genotype.

## Cell culture

Primary cortical neuronal cultures, derived from E16.5 wild-type C57BL/6J fetuses, were plated in 10% serum-containing neuronal culture medium (neurobasal medium containing B27 supplements, penicillin, streptomycin, and GlutaMax) on either coverslips or culture dishes that were pre-coated with poly-L-ornithine (Sigma p4957). The medium was changed 4 h after the initial plating to serum-free neuronal culture medium, and then a half-medium change was performed every three days. The SH-SY5Y neural cell line, as well as the WT and REST KO mouse embryonic fibroblasts (MEFs), were maintained in Dulbecco's Modified Eagle Medium (DMEM) with 4.5 mg l$^{-1}$ D-glucose, 2 mM L-glutamine, 1 mM sodium pyruvate, 100 U ml$^{-1}$ penicillin, 100 μg ml$^{-1}$ streptomycin (all from Invitrogen), supplemented with 10% FBS. Cells reaching a 80–90% confluency were split after digestion with Trypsin 0.25% and plated in DMEM with 4.5 mg l$^{-1}$ D-glucose, 2 mM L-glutamine, 1 mM sodium pyruvate, 100 U ml$^{-1}$ penicillin, 100 μg ml$^{-1}$ streptomycin, supplemented with 10% FBS.

Primary cortical neurons were treated with the following drugs for 24 h: human recombinant DKK-1 (250 ng/mL), XAV939 (2.5 μM), ICG001 (5 μM), GSK2606414 (0.5 μM), Thapsigargin (TG; 1 μM), CHIR99021 (Chiron; 1 μM), lithium chloride (1 mM), or combinations of TG (1 μM) and Chiron or lithium chloride (1 mM).

## Immunocytochemical analysis of cultured cells

Cells plated on poly-L-ornithine-coated coverslips were fixed by incubation with 4% (v/v) paraformaldehyde in PBS for 20 min at room temperature, followed by three washes in PBS, and then permeabilized with 0.1% Triton X-100 in PBS for 15 min at room temperature. After 3 additional washes in PBS, cells were blocked with 3% BSA and 0.05% Tritonx100 in PBS for 45 min at room temperature. Primary antibodies were diluted to the appropriate concentration in 3% BSA and 0.05% Tritonx100 in PBS and incubated overnight at 4 °C. Cells were then washed 3 times with PBS for 10 min before adding fluorophore-conjugated secondary antibodies for 2 h at room temperature. Fluorophore-labeled cells were then washed in PBS for 3 × 10 min, and then mounted using Prolong Gold® mounting medium with DAPI and anti-fade reagent (Invitrogen).

## PCR

To extract RNA, the TRIzol kit (Invitrogen) was used for both cells and tissues. All RNAs were DNAse treated. Primers were obtained from the Harvard's PrimerBank (https://pga.mgh.harvard.edu/primerbank/). The PrimerBank contains over 306,800 primers covering most known human and mouse genes. The primers used for real time PCR in this study are listed in Supplementary Data 6. Real-time PCRs were run for 40 cycles. The purity of both PCR and RT–PCR products were determined by single peak melting curves.

## Generation of REST-deficient mouse embryonic fibroblasts

We crossed REST$^{lx/lx}$ and REST$^{lx/lx}$ mice to generate REST$^{lx/lx}$ embryos. Pregnant REST$^{lx/lx}$ females were sacrificed at embryonic day 13.5 (E13.5) and the embryos were dissected. The head, liver and heart were removed, and the remaining tissue was minced and then digested with 1 ml of 0.05% trypsin/EDTA (Gibco, Invitrogen), including 100 Kunitz units of DNase I (USB) per embryo. Cells were pooled from both male and female embryos (1:1 ratio) and were then placed in warm culture medium containing DMEM, 10% FBS, 2 mM L-glutamine and 1% penicillin-streptomycin. The cells were plated in a flask coated with 0.2% type B gelatin (Sigma) for 2 h. The fibroblasts (P0, passage 0) are the only cells that have the ability to attach to the gelatin-coated flasks. When the cells became 80–90% confluent, they were further expanded (passages 1–3). MEFs were transduced with a retroviral vector carrying nlsCre-Hyg, allowing expression of nuclear-targeted Cre and hygromycin selection. The infected MEFs were passaged multiple times under stringent hygromycin selection, to ensure selection of cells with efficient integration and stable Cre expression. Cre-mediated REST deletion in REST$^{lx/lx}$ MEFs was confirmed by PCR amplification of the Cre-recombined REST allele (REST$^{rec}$). Loss of REST expression in REST KO MEFs was confirmed by WB and immunofluorescence labeling of fixed cells.

## MEF transfection and Aβ measurement

WT and REST KO mouse embryonic fibroblasts (MEFs), as well as REST KO MEFs that had previously been infected with hREST-IRES-GFP-expressing lentiviruses, were cultured at 80–90% confluency in 12-well plates and 1 μg of either human APP$^{WT}$ or human APP$^{Swe}$-carrying pcDNA3.1 plasmid was mixed with Lipofectamine 3000 reagent (Invitrogen) in OptiMEM medium (Invitrogen) and added to each well. 48 h after lipofection, the medium and the cells were harvested. The concentrations of Aβ40 and Aβ42 in the cell culture medium were determined using human Aβ40 (Wako, Cat. No. 292-62301) and Aβ42 (Wako,

Cat. No. 298-62401) ELISA kits, according to the manufacturer's instructions.

## Generation of recombinant lentiviruses for *REST* knockdown and overexpression

The REST shRNA (sh4) and control shRNA, as well as REST over-expressing and control lentiviruses have been previously described[13]. Briefly, to achieve REST knockdown in the neural SH-SY5Y cell line, short-hairpin RNAs against human REST were expressed from the lentiviral plasmid pGIPZ (Open Biosystems). Mature antisense sequences of REST shRNAs are TTTGAACTGTAAATATCTG (392–410) (shREST-4 or sh4). Numbers in parenthesis indicate the position of mature antisense sequences corresponding to the coding domain sequence (CDS) of human *REST* (3294 bp). As a control, the GIPZ non-silencing shRNAmir was used (Open Biosystems). For REST over-expression, human REST-IRES-GFP- or control GFP-expressing expressing lentiviruses were generated in HEK293T cells by co-transfection of transfer vectors, packaging plasmid (pRSV-REV and pMDLg) and envelope plasmid (pCMV-VSVG) using Lipofectamine 2000 (Invitrogen)[13]. Viruses were titrated by infecting SH-SY5Y cells at serial dilution. We determined that a multiplicity of infection (MOI) of 10 can efficiently transduce >90% of SHY-5Y neuroblastoma or MEF cells. 500,000 cells/mL were infected with lentiviruses at a MOI of 10. 72 h after infection, cells were harvested, and the RNA was purified.

For in vivo delivery of the human *REST* gene in the mouse hippocampus, full length human *REST* cDNA was amplified by PCR with primers CQO230 (CCgctagcATGGCCACCCAGGTAATGGGG) and CQO231 (GGgaattcttaCTCCTGCCCTTGAGCTGCTTC), and subcloned into an pAAV-CW3SL vector at the NheI/EcoRI sites. The human *REST* cDNA was placed under the control of the CaMKIIa promoter and a shortened expression cassette, to fit the 4.8-kb AAV packaging size limit. Shortened WPRE3 and SV40 late Poly(A) sequences in the pAAV-CW3SL vector were as described[45]. GFP cloned in the same pAAV-CW3SL vector was used as a control. AAV9 were packaged at Boston Children's Hospital's viral core, where viral titers were determined by qPCR.

## Membrane preparation and extraction for γ-secretase assessment

CHAPSO-solubilized membranes were prepared from WT and REST knock-out MEF cells. 10 confluent 15-cm plates were scraped and lysed in buffer containing 50 mM MES, pH 6.0, 150 mM NaCl, 5 mM MgCl$_2$, 5 mM CaCl$_2$, and Roche complete protease inhibitors by three passages through French pressure cell at 1000 p.s.i. The lysate was first spun at low speed and then at $100,000 \times g$. The resulting membrane pellet was resuspended in 100 mM sodium bicarbonate buffer, pH 11.3, and spun again at $100,000 \times g$ to remove peripheral membrane-associated proteins. The washed membranes were then resuspended in 80 µL of Hepes buffer, pH 7.0, containing 1% CHAPSO, incubated for 1 h on ice, and spun at $100,000 \times g$. The CHAPSO-soluble supernatant was then collected, and the protein concentration in each solubilized membrane preparation was determined by BCA assay (Pierce).

## Detection of γ-secretase components

Samples of CHAPSO-solubilized membranes from REST and KO MEFs were prepared in SDS sample buffer and incubated at room temperature for 10 min. Each sample contained 2 µg of total protein based on the protein concentrations measured by BCA assay. The proteins were resolved using 12% Bis Tris criterion gels (Bio-Rad) and transferred to PVDF, followed by western blotting to detect γ-secretase components and transferrin receptor (as a loading control). Signal was captured using ECL, and the levels of γ-secretase components and transferrin receptor were quantified by densitometry using Image-Quant (GE Healthcare).

## γ-Secretase enzymatic assay

Solubilized membranes from WT and REST KO MEFs were incubated with 2 µM C100-FLAG substrate in HEPES buffer at pH 7.0 with 0.1% phosphatidylcholine, 0.025% phosphatidylethanolamine, 0.00625% cholesterol, and a final CHAPSO concentration of 0.25% for 1 h at 37 °C. Equal amounts of solubilized protein were added to each reaction based on the protein concentration determined by BCA assay. AICD-FLAG product was detected by incubating equal volumes of WT and REST KO reactions in SDS sample buffer for 10 min at room temperature. The reaction samples were run on 12% Bis Tris Criterion gels (Bio-Rad), followed by transfer of proteins to PVDF, western blotting with M2 anti-FLAG antibody (Sigma), and exposure of the blot using ECL (GE Healthcare). AICD bands were then quantified by densitometry using ImageQuant software (GE Healthcare).

## Western blotting

Mouse brain samples (20-30 mg) were homogenized in 20 volumes of T-PER buffer (Invitrogen) supplemented with Complete® protease inhibitors and PhosStop® phosphatase inhibitors. Samples were centrifuged for 15 min at $22,136 \times g$ at 4 °C and the supernatant was collected. Protein concentrations were determined using the Micro BCA Protein Assay kit (ThermoScientific, Cat.No. 23235). Laemmli 4x buffer containing β-mercaptoethanol was added to samples and the samples were boiled at 90 °C for 10 min then ran on SDS-PAGE gels. Gels were transferred onto PVDF or nitrocellulose membranes. Membranes were blocked with 5% non-fat dry milk powder dissolved in TBS-T 1x (Tris buffered saline containing 0.05% Tween 20) for 45 min at room temperature. Primary antibodies were diluted in 5% non-fat dry milk powder dissolved in TBS-T 1x and were incubated with the membranes overnight at 4 °C. Membranes were washed for 3 × 10 min in TBS-T 1x, then secondary antibodies, diluted in 5% non-fat dry milk powder dissolved in TBS-T 1x, were added for 1 h at room temperature. After three 10-min washes in TBS-T 1X, the membranes were incubated with Western Blotting detection Reagents (Amersham, RPN2209).

## Chromatin immunoprecipitation

Four biological replicates, each comprised of $n = 4$ pooled frozen cortices from 3xTg mice (females, age 11 months) were processed for ChIP-seq (total $n = 16$ 3xTg cortices). Similarly, four biological replicates, each comprised of $n = 4$ pooled frozen cortices from WT mice (females, age 11 months) were processed for ChIP-seq (total $n = 16$ WT cortices). The frozen tissue was sent to Active Motif Services (Carlsbad, CA) to be processed for ChIP-Seq. In brief, the tissue was submersed in PBS + 1% formaldehyde, cut into small pieces and incubated at room temperature for 15 min. Fixation was stopped by the addition of 0.125 M glycine (final). The tissue pieces were then treated with a TissueTearer and finally spun down and washed 2x in PBS. Chromatin was isolated by the addition of lysis buffer, followed by disruption with a Dounce homogenizer. Lysates were sonicated and the DNA sheared to an average length of 300–500 bp. Genomic DNA (Input) was prepared by treating aliquots of chromatin with RNase, proteinase K and heat for de-crosslinking, followed by ethanol precipitation. Pellets were resuspended and the resulting DNA was quantified on a NanoDrop spectrophotometer. Extrapolation to the original chromatin volume allowed quantitation of the total chromatin yield.

An aliquot of chromatin (25 ug) was precleared with protein A agarose beads (Invitrogen). Genomic DNA regions of interest were isolated using 4 ug of antibody against the C-terminal region of REST (generous gift from Gail Mandel[21,37], Vollum Institute). Complexes were washed, eluted from the beads with SDS buffer, and subjected to RNase and proteinase K treatment. Crosslinks were reversed by incubation overnight at 65 °C, and ChIP DNA was purified by phenol-chloroform extraction and ethanol precipitation.

## ChIP sequencing

Illumina sequencing libraries were prepared from the ChIP and Input DNAs by the standard consecutive enzymatic steps of end-polishing, dA-addition, and adapter ligation. Steps were performed on an automated system (Apollo 342, Wafergen Biosystems/Takara). After a final PCR amplification step, the resulting DNA libraries were quantified and sequenced on Illumina's NextSeq 500 (75 nt reads, single end). Reads were aligned to the mouse genome (mm10) using the STAR algorithm[46] (v2.7.0a) (--alignIntronMax 1 --alignEndsType EndToEnd). Peak locations were determined using the MACS algorithm[47] (v2.1.0) (--nomodel) and normalized fold enrichment tracks were generated by using the callpeak function with --SPMR, then passing the bedgraph outputs into the bdgcmp function with the setting -m FE (fold enrichment). Bedgraph files were converted into bigWig and visualized using Integrated Genome Viewer (IGV, v2.4). Peaks were filtered for fold enrichment more than 5 and q-value less than 0.01; furthermore, peaks that were on the ENCODE blacklist of known false ChIP-seq peaks were removed. This analysis uncovered $n = 3006$ peaks in WT cortex and $n = 5077$ in 3xTg (Supplementary Data 3). Motif enrichment analysis was performed using HOMER[48]. *Bedtools Intersect* was used to overlap peaks from WT and 3xTg samples to determine common regions in each group; this uncovered $n = 2503$ peaks that were shared bweteen WT and 3xTg (Supplementary Data 4). Differential binding sites was identified using *DiffBind* (https://bioconductor.org/packages/devel/bioc/vignettes/DiffBind/inst/doc/DiffBind.pdf) using default settings (the results are shown in Supplementary Data 5). Peaks were annotated using ChIPseeker[49], and the genes located near the peak (transcription start site up to 5 kb upstream or 5 kb downstream) were identified. *computeMatrix* (v3.3.2) was applied to build a matrix with a flanking region of ±1 kb around the center of enriched REST peaks and *plotHeatmap* (v3.3.2) was used to generate heatmaps. We employed DAVID for functional enrichment analysis and enrichment of gene ontology (GO) terms[50,51] was calculated using Fisher's exact test with Bonferroni correction. ChIP-seq data has been deposited at the Gene Expression Omnibus (GEO): ncbi.nlm.nih.gov/geo/query/acc.cgi?acc=GSE195446 (accession code GSE195446).

## REST ChIP-qPCR

ChIP assays were performed as previously described[13] with minor modifications. Briefly, mouse cortices were cross-linked using 1% formaldehyde at room temperature for 10 min. The reaction was stopped by addition of 125 mM glycine. Brain tissue was washed with cold PBS and resuspended in 1 ml cell lysis buffer (0.5% Triton, 85 mM KCl, 15 mM NaCl, 4 mM MgCl2, 5 mM HEPES, pH 7.6, 0.5 mM DTT, protease inhibitor cocktail). Tissue was homogenized using a dounce homogenizer (pestle B, 15 strokes) and the homogenate was passed through a 40 μm strainer. Samples were centrifuged at $4000 \times g$ at 4 °C for 5 min to isolate nuclei, which were resuspended in nuclear lysis buffer (50 mM Tris pH 8, 150 mM NaCl, 2 mM EDTA, 1% Triton, 1% SDS, 0.1% sodium deoxycholate, 0.5 mM DTT, protease inhibitor cocktail). Genomic DNA was sheared by sonication (Bioruptor) at high intensity (30 s ON/OFF, 10 cycles) to generate 300–1000 bp fragments. The sonicated samples were centrifuged for 10 min at $15,000 \times g$ at 4 °C. The supernatant was diluted 1:5 in ChIP buffer (0.1% SDS, 1% Triton X-100, 2 mM EDTA, 140 mM NaCl, 50 mM Tris-HCl, pH 8) and precleared with 10 μl Protein A agarose beads (Life Technologies) that was equilibrated with 1% BSA. 2 μl of REST antibody, or non-specific IgG control antibody (Sigma, Catalog No. 17-641) was added to 20 μl of Protein A agarose beads in ChIP buffer and incubated for 2 h at room temperature to obtain REST-conjugated beads. 5% of the pre-cleared DNA was reserved as input control, and the remaining sample was incubated with REST-conjugated beads overnight at 4 °C. Beads were washed twice with low salt wash buffer, once with high salt wash buffer, once with LiCl wash buffer and twice with TE buffer. The beads were then incubated with elution buffer (1% SDS, 0.1 M NaHCO3) for 30 min at

37 °C. Reverse-crosslinking was performed at 65 °C overnight with 200 mM NaCl. DNA was further treated with RNase A and proteinase K and purified using the QIAGEN PCR purification kit. Quantitative PCR (qPCR) reactions on ChIP chromatin were carried out in triplicate on specific genomic regions using SYBR Green Supermix (KAPA Biosystem). The resulting signals were normalized for primer efficiency by carrying out qPCR for each primer pair using input DNA. qPCR was performed using primers summarized in Supplementary Data 6.

## Stereotaxic injection of AAV9-REST virus

Stereotaxic surgery was performed as previously described[52] with modifications below. Briefly, 3xTg, 3xTg;GT or J20 mice at the indicated ages were anesthetized with Ketamine/Xylazine (80 mg/kg Ketamine and 10 mg/kg Xylazine by intraperitoneal injection) and placed on the stereotaxic instrument with a heating pad to maintain body temperature. We bilaterally injected 500 nL ($3 \times 10^8$ vg total) of AAV9-hREST or AAV9-GFP control using a Hamilton syringe at a speed of 100 nL/min, to minimize loss of material and physical damage to the brain. The bilateral injection sites were −2 mm posterior, ±2 mm lateral, and −1.5 mm ventral from the bregma. The mice were kept on a heating pad to maintain the body temperature at 37 °C throughout the procedure and until they were awake and recovered. The activity of injected mice was carefully monitored daily for a week and twice weekly afterwards. The suture was removed 10 days after the surgery. Mice were sacrificed 10 weeks after the injection, for cardiac perfusion and brain collection as described above. The brains were collected and subjected to pathological evaluation of tau and Aβ, analyses of REST protein expression, and mRNA expression of *App* and *Tau* (*Mapt*) transgenes and selected REST targets.

## Statistical analysis

Statistical analysis was performed using the GraphPad software (version 9.3.1). Statistical tests used are noted in the figure legends or in the relevant Methods section. For comparisons between two groups, we used parametric two-tailed *t*-tests for normal distributions and non-parametric two-tailed Mann–Whitney U tests for non-Gaussian distributions. For comparisons between 3 or more groups, we used ordinary one-way ANOVA folowed by Tukey's multiple comparisons test for Gaussian distributions, and Kruskal–Wallis one-way ANOVA followed by Dunn's multiple comparisons test for non-Gaussian distrubutions. To assess the interaction of two variables (such as the interaction between sex and genotype) and their effects on the parameters we investigated, we conducted two-way ANOVA analyses. Throughout the paper, all tests are two sided and unpaired unless stated otherwise. A significance level of 0.05 was used to reject the null hypothesis. The summary of each statistical test, including information on sample sizes, gender, confidence intervals and *P*-values can be found in Supplementary Data 1.

## Reporting summary

Further information on research design is available in the Nature Portfolio Reporting Summary linked to this article.

## Data availability

Source data are provided with this paper. Data is available upon request. Clinico-pathological data on post-mortem human samples from ROSMPAP can be requested at https://www.radc.rush.edu. ChIP-seq data has been deposited at Gene Expression Omnibus (GEO): ncbi.nlm.nih.gov/geo/query/acc.cgi?acc=GSE195446 (accession code GSE195446). Source data are provided with this paper.

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

## Acknowledgements

We thank members of the Yankner laboratory for suggestions and discussion. We also thank Miglia Cornejo, Allison Harwick and Saba Hazaveh for technical assisstance; Sebastian Valentin and Andrew Thompson for assistance with tissue sectioning, and Greg Klein for assistance with brain sample selection. This work was supported by an NIH Director's Pioneer Award (DP1OD006849) and NIH grants RO1AG046174 and RO1MH113279 to B.A.Y., K01AG051791 and DP2AG072437 to E.A.L., P30AG10161, R01AG15819, R01AG17917, and U01AG61356 to D.A.B., RO1AG66986 to M.S.W., and grants from the Ludwig Family Foundation and the Glenn Foundation for Medical Research to B.A.Y.

## Author contributions

L.A. performed and analyzed experiments involving the generation and characterization of REST knockout mice, human neuropathology, and cell culture. C.Q performed and analyzed the AAV9-REST overexpression experiments. Z.K.N performed and analyzed REST ChIP-seq and ChIP-PCR experiments. M.L., P.R., E.L.B., E.K.L., S.E.H. and M.Y. performed and analyzed REST knockout mouse and cell culture experiments. D.D., J.C. and E.A.L. performed bioinformatic analysis. M.A.F and M.S.W. performed experiments on gamma secretase. D.A.B. contributed human brain samples and clinical data. L.A. and B.A.Y. wrote the manuscript with input from C.Q., Z.K.N., J.C., M.A.F., M.S.W., D.A.B. and E.A.L. B.A.Y. supervised all aspects of the work.

## Competing interests

The authors declare no competing interests.
