## [Peer Review File · Nature Communications]

A Neurodegeneration Checkpoint Protects Against the Onset of Alzheimer's DiseaseREVIEWER COMMENTS

Reviewer #1 (Remarks to the Author):

In this study, Aron et al. performed a series of analyses (i.e. immunofluorescence, IHC, WB, CHIP-seq) in human postmortem brain tissue, mouse and cell culture models of AD to investigate the role of the transcriptional repressor REST in Alzheimer's disease (AD). The authors concluded that activation of REST inhibits the progression of AD pathology, as well as cognitive decline and neurodegeneration. Based on REST CHIP-seq data analysis in the 3xTg mouse model vs wild-type, the authors propose a mechanism through which REST blocks disease spreading by downregulating genes involved in the generation of A β and tau phosphorylation (i.e. γ -secretase and tau kinase genes) as well as genes involved in other pathways (cell death and inflammation). This work is based on a previous study (Lu et al., Nature, 2014) from the same group.

While functional experiments in mice and in cell culture models show an effect of loss of REST in neurodegeneration-related processes, the CHIP-seq data analyses of REST binding in the 3xTg mouse model require further clarifications:

- The authors performed REST CHIP-seq in 2 biological replicates from 3xTg and wild type mice (4 pooled frozen cortices per study group), but no statistical analyses were performed to identify sites with significant changes in REST binding between the 3xTg and wild type mice. If no statistically significant differences in REST binding were identified, the data cannot support the claims above. The authors could repeat the experiments with more than 2 biological replicates without pooling brains from each mouse if sample size is the issue.
- It is not clear how the gene targets of REST were identified and what the overall distribution of REST in the genome is. Examples of UCSC genome browser tracks showing REST binding at genes involved in AD should be shown. Also, a link to the CHIP-seq data visualization through the UCSC genome browser (or similar) should be provided. The chromosome view in Extended Data Fig. 8 doesn't allow to distinguish background from specific (peak) REST binding.
- It is concerning that only 20-30% of the CHIP-seq reads could be used for downstream data analyses. A table showing sequencing and alignment statistics (total reads, total aligned reads, uniquely aligned reads, etc.) should be provided.
- RNA-seq should be performed together with CHIP-seq in the 3xTg mice to show a negative correlation between REST binding (repressor) and gene expression and repression of the genes targeted by REST, especially those involved in AD pathology.
- Ideally, given the availability of postmortem brain samples, REST CHIP-seq should be performed in human AD brains as epigenomic changes in mouse models don't definitively recapitulate those in human.

Reviewer #2 (Remarks to the Author):

Using a combination of molecular signaling, imaging, and behavioral tests on transgenic mice, and molecular signaling and imaging in human tissues, the authors demonstrate that REST serves as a neurodegeneration checkpoint with multiple molecular targets that protects against the onset of AD. The authors used a combination of genetic AD models and tested the role of REST in cognitive deficits by knockout or knockdown REST expression. By examining nuclear REST abundance in PFC (prefrontal cortex) of aging individuals with NCI or clinically diagnosed AD, the authors demonstrate that REST is induced in cognitively-intact aging AD mouse models. By inactivating REST by cKO or REST gene trap, the author further demonstrates that REST inactivation accelerates A β deposition and pathogenic tau accumulation in 3xTg mice and J20 mice. Accelerated A β deposition and tau accumulation were found in AD mice carrying a second REST loss-of-function allele. In the J20 AD mice with or without REST gene trap, the author demonstrated that loss of REST is required for A β -induced phospho-tau accumulation. Moreover, REST nuclear abundance is relevant to the cognitive decline and neurodegeneration during aging in 3xTg;cKO and J20 mice. By using CHIP-seq analysis, the authors reveal the presence of different REST targets in WT and 3xTg mice. Knockdown of REST is associated with elevated expressions of components of gamma secretase and γ increased generation of A β 40 and A β 42 in vitro.

In a previous paper (Nature 2014), many of the same authors showed that during normal aging, REST is induced and is degraded together with pathological misfolded proteins. The conclusion is similar to that of the current manuscript that REST protects neurons from AD pathology during aging. The present paper would be significantly strengthened were the author to reveal the mechanism by which REST is induced in the early stages of AD mouse models and how REST can protect neurons.

The authors do not sufficiently defend their conclusion that REST serves as a neurodegeneration checkpoint during aging. Most of the results are correlative. Although they use several mouse models of AD and demonstrate that loss of REST causes AD pathology and a decline in cognition, they do not examine whether overexpression of REST could reduce AD pathology or cognitive decline. Some of the MWM data are inconsistent with those obtained with other behavioral paradigms. Authors should at the least discuss this apparent discrepancy in the Discussion section.

Specific comments:

1, Fig 1, the authors show nuclear REST levels were significantly reduced in AD cases relative to NCI cases. However, in Fig 2, their results indicate that more REST appears in the nucleus of neurons that exhibit misfolded and phosphorylated tau in 3xTg AD mouse model. Can they resolve this apparent discrepancy?

2, Fig 4, line 255, "Fig. 4e,f" should be "Fig. 4d,e". And there is no J20: cHET group in the learning latency of MWM in Fig. 4d. But the authors still showed the number of entries and time spent in target area of this group in Fig. 4e.

3, Extended Data Fig. 7i and j: the results show that swimming speed of all three J20 mice groups is reduced compared with WT mice. However, in panel i, mice in all these three groups traveled

significantly longer distances than did WT mice.

4, Extended Data Fig. 9, the authors show that REST targets 555 genes in 3xTg mice (mouse brain), but 663 genes in WT mice. Thus, REST targets 115 genes exclusively in the 3xTg, but not WT, mice. To prove that not only REST, but also a host of REST target genes, are relevant to AD pathology, the authors need to provide (or at least discuss) mechanisms by which the target genes might cause neuronal death.

Reviewer #3 (Remarks to the Author):

Amyloid and tau pathology can be profound in brains of some aged individuals with no cognitive deficits. Investigation of the underpinning mechanisms in this specific population could provide us important insightful mechanism, herein proposed as a mechanism of resilience. This study proposed a neurodegeneration checkpoint response manifested by REST to early pathological changes in AD that restricts further disease progression. One of these responses is to suppress development of both amyloid and tau pathology. Using a well-stratified human brain samples from the ROSMAP cohort, the authors first showed by IHC a pathology-dependent increase of nuclei REST level in PFC neurons in NCI brain samples, which was not detected in AD samples. This observation was further replicated in brains and primary neuronal cultures of aged 3xTg mice. To study the consequence of REST activation in NCI brain, they deleted REST in CA1 glutamatergic hippocampal neurons, in cortical/hip glutamatergic neurons or a het REST null allele in 3xTG with tau and amyloid or in J20 mice with amyloid only. Strikingly, they found that REST deletion substantially increased both amyloid and tau pathology, worsen cognitive deficits and increase neuronal death in both models. By performing ChIP-seq, they identified REST-targeted genes in WT and 3xTG mice. Interesting, they found some AD-pathology related genes targeted by REST, including tau kinases (such as Cdk5 and Gsk3beta) and g-secretase component PEN2. With validation of these genes in mouse models as well as in MEF cells with REST knockout or overexpression, they proposed that brain pathology induces REST activation and alter transcriptomic profile targeted by REST in AD brains, resulting in reduction of A β deposition and tau accumulation, and potentially downregulate other responses including immune responses and cell death pathways. This is a very intriguing discovery revealing an unknown mechanism how neurons response to pathology at early stage by establishing a checkpoint response to prevent further development of pathology. This finding is significant, suggesting that activating neuronal REST could lead to an effective tool for AD treatment. The experimental approaches using multiple genetically modified mouse lines are rigorous and powerful. The data provide strong support of their conclusion. Considering some concerns listed below, a revised version is recommended before acceptance for publication in this journal.

Major concern:

J20 mice with massive amyloid deposition in brain have cognitive deficits in spatial learning as shown in MWM test even though REST level is increased in neurons. This is clearly different from human NCI individuals. How to justify this model in this study?

Although the phenotype on amyloid and tau pathology by REST deletion is striking, a rescue experiment will be helpful to solidate these findings. This can be achieved by viral-mediated REST overexpression in CA1 pyramidal neurons in REST cKO mice with J20 or 3xTg background.

The sources for multiple mouse lines (where these mice were obtained) are missing in the Method, including 3xTg, J20, CA1-specific CamKII α -Cre or forebrain-specific CamKII α -Cre mice. Considering the fact that 3xTg is usually used as homozygote by this field, it is important to provide the rationale of using 3xTG het and clearly describe the breeding strategy for 3xTg with various REST mice.

Minor concerns:

1. Figure 1a Although the extended figure 1c shows nuclei location of REST, it is necessary to include DAPI channel in here. It seems that pTau staining in NCI early pathology samples is mainly in nuclei, which is weird. Has this been reported before? An explain or discussion of this phenotype will be helpful.
2. Figure 1e: The amyloid pathology in 3-month-old J20 mice need to be shown.
3. "Independent analysis did not show a significant effect of gender on the parameters described below (Extended Data 7)." It is not clear where to locate this data. Do you mean Extended Data table 7?
4. Figure 2a: DAPI channel need to be shown.
5. Figure 3a,b: n=3 for REST cKO-3xTg has limited power to support the conclusion. Although the REST level is examined in WT and KO, it is still necessary to show the levels of REST in 3xTg background by IHC or western blot considering the pathology induces REST.
6. Extended data Figure 3: n=3 embryos/group, were these done in one experiment or three independent experiments? If done in one experiment, this needs to be repeated three times.
7. Figure 4d was not mentioned or discussed in the Result section. Figure 4f does not exist.
8. Figure 5 and Extended figure 9: what are the color codes in these graphs?

REVIEWER COMMENTS

We thank the reviewers for their constructive comments and suggestions which have improved the paper. We believe we have addressed all the major concerns with additional experiments and analysis. The paper now provides a novel conceptual framework for the role of REST in the prevention and progression of Alzheimer's disease. The revised manuscript includes new mechanistic data on signaling pathways that activate REST in response to AD pathology, more in-depth ChIP-seq analysis, and exciting new findings on the therapeutic effects of REST overexpression in the brain. The responses to specific reviewer comments are bolded below, and major new sections of text are highlighted in the manuscript.

Reviewer #1:

In this study, Aron et al. performed a series of analyses (i.e. immunofluorescence, IHC, WB, ChIP-seq) in human postmortem brain tissue, mouse and cell culture models of AD to investigate the role of the transcriptional repressor REST in Alzheimer's disease (AD). The authors concluded that activation of REST inhibits the progression of AD pathology, as well as cognitive decline and neurodegeneration. Based on REST ChIP-seq data analysis in the 3xTg mouse model vs wild-type, the authors propose a mechanism through which REST blocks disease spreading by downregulating genes involved in the generation of A β and tau phosphorylation (i.e. γ -secretase and tau kinase genes) as well as genes involved in other pathways (cell death and inflammation). This work is based on a previous study (Lu et al., Nature, 2014) from the same group.

While functional experiments in mice and in cell culture models show an effect of loss of REST in neurodegeneration-related processes, the ChIP-seq data analyses of REST binding in the 3xTg mouse model require further clarifications:

We have revised and expanded the presentation of the ChIP-seq data in Fig. 4 to increase clarity and convey more information. First, peaks were identified using the MACs algorithm (default cut-off p-value 1e-7). Peaks were further filtered by removing false positive ChIP-seq peaks using the ENCODE blacklist. Overlapping peaks between biological replicates were combined and merged into a final peak list as previously described (Grubert et al., 2020. Nature 583, 737–743). We identified gene targets that were predominantly bound by REST in 3xTg or wild-type mice, as well as targets common to both, which is now illustrated in a Venn diagram (Fig. 4a and Extended Data Fig. 4a). Importantly, analysis of known and *de novo* transcription factor binding motifs shows that the REST-RE1 binding motif is the most enriched sequence found in ChIP-seq peaks in both 3xTg and WT mice, indicating specificity (Fig. 4d). Interestingly, gene targets enriched in 3xTg are involved in kinase activity, cell death, stress responses and the unfolded protein

response (Fig. 4c). These results suggest that there is a dynamic shift in REST gene targeting associated with early pathology in 3xTg mice.

1. The authors performed REST ChIP-seq in 2 biological replicates from 3xTg and wild type mice (4 pooled frozen cortices per study group), but no statistical analyses were performed to identify sites with significant changes in REST binding between the 3xTg and wild type mice. If no statistically significant differences in REST binding were identified, the data cannot support the claims above. The authors could repeat the experiments with more than 2 biological replicates without pooling brains from each mouse if sample size is the issue.

The unbiased ChIP-seq was performed on 2 biological replicates (4 pooled cortices/replicate) for both WT and 3xTg mice. Such a design, aimed at discovering genome-wide binding sites, is used routinely in the field (e.g. Grubert et al 2020. Nature 583, 737–743; Partridge et al. 2020. Nature 583, 720–728). As suggested by the reviewer, we have now performed ChIP analysis on individual WT and 3xTg brain samples followed by qPCR of selected genomic regions based on REST binding sites resolved in the unbiased ChIP-seq of pooled samples. ChIP-qPCR of individual samples confirmed REST binding to genes previously resolved by ChIP-seq that contribute to AD pathology, namely the major tau kinases CDK5 and GSK3 β . The ChIP-qPCR analysis confirmed elevated REST binding to these genes in 3xTg versus wild-type mice (Fig. 4 e,f). This was also observed for the inflammation-related pro-apoptotic gene *DAXX*. The ChIP-qPCR experiments were subjected to rigorous statistical analysis which showed that the elevated REST binding to these genes in 3xTg mice was highly significant in 5-month-old and 11-month-old 3xTg mice, suggesting that the REST-mediated checkpoint is both robust and sustained. For each gene targeted by REST, we included a negative control site 10 kb downstream from the REST binding site; these control sites did not show significant REST binding. Furthermore, we also validated REST binding to the promoter of the p53 gene (*Trp53*), a major pro-apoptotic regulator, in both WT and 3xTg samples (Fig. 4 e,f). Finally, two positive control genes, *Tle3* and *Aes*, showed similar REST binding in 3xTg and wild-type, whereas the negative control site *Untr6* showed no significant REST binding (Extended Data Fig. 4e). Taken together, the unbiased ChIP-seq on pooled mouse samples and the quantitative ChIP-PCR on individual mice suggest that elevated REST binding occurs in a subset of functionally important target genes in the setting of early AD pathology. Moreover, we show that the corresponding proteins were significantly upregulated in REST-deficient 3xTg mice (3xTg;cKO and GT) relative to 3xTg on a wild-type REST background (Fig. 5a-d). These results suggest that increased REST binding to these genes represses their expression.

2. It is not clear how the gene targets of REST were identified and what the overall distribution of REST in the genome is. Examples of UCSC genome browser tracks

showing REST binding at genes involved in AD should be shown. Also, a link to the ChIP-seq data visualization through the UCSC genome browser (or similar) should be provided. The chromosome view in Extended Data Fig. 8 doesn't allow to distinguish background from specific (peak) REST binding.

Please see the initial response above for the explanation of how gene targets of REST were identified. We now provide examples of UCSC genome browser tracks showing REST binding at genes involved in AD in both 3xTg and wild-type mice. Moreover, the new Fig. 4 provides a more in-depth analysis and visualization of REST binding sites in the genome. The link to the ChIP-seq data visualization has also been provided (please see the Reporting Summary PDF file that accompanies the manuscript files). The ChIP-seq data files can be accessed at: ncbi.nlm.nih.gov/geo/query/acc.cgi?acc=GSE195446

3. It is concerning that only 20-30% of the ChIP-seq reads could be used for downstream data analyses. A table showing sequencing and alignment statistics (total reads, total aligned reads, uniquely aligned reads, etc.) should be provided.

We would like to clarify that 72-93% of the ChIP-seq reads were uniquely mapped in our samples (all above 29 million reads) and used for downstream analysis, which is well within ENCODE ChIP-seq guidelines. As requested by the reviewer, we now provide a table showing the sequencing and alignment statistics (Extended Data Table 2).

4. RNA-seq should be performed together with ChIP-seq in the 3xTg mice to show a negative correlation between REST binding (repressor) and gene expression and repression of the genes targeted by REST, especially those involved in AD pathology.

A challenge of using bulk RNA-seq to examine the selective induction of REST in excitatory neurons is the dilution effect of cells in which REST is not induced in 3xTg mice such as glial cells. To assess the regulation of specific AD-related REST targets in neurons, high resolution immunofluorescence microscopy was performed with rigorous quantitative analysis and controls. REST-deficient 3xTg mice were compared to 3xTg mice on a normal REST background to assess expression changes directly related to REST function. We show that CDK5, GSK3 β and PEN-2 (gamma secretase subunit) levels were significantly elevated in cortical and hippocampal neurons in REST-deficient 3xTg mice (independent cohorts of 3xTg;GT and 3xTg;cKO) relative to 3xTg mice on a wild-type REST background (Fig. 5a-d and Extended Data Fig. 6a, b). This was corroborated by Western blot analysis, which showed significantly elevated CDK5 and GSK3 β levels in REST-deficient 3xTg;GT mice relative to 3xTg mice on a wild-type REST background (Fig. 5e). Finally, we show that *Cdk5*, *Gsk3 β* and *Daxx* mRNA expression is significantly elevated in REST-deficient 3xTg:GT mice, and that this is rescued by overexpressing REST with an AAV vector (Fig. 9d). These findings complement the pathology of the REST-deficient AD mouse models, which is

characterized by highly significant elevation of phospho-tau epitopes and A β accumulation.

5. Ideally, given the availability of postmortem brain samples, REST ChIP-seq should be performed in human AD brains as epigenomic changes in mouse models don't definitively recapitulate those in human.

This would be a valuable analysis, however ChIP-seq has proven to be difficult in the human AD brain, presumably because large quantities of aggregated protein confound the chromatin immunoprecipitation. We have attempted this but the quality of the data was poor with substantial nonspecific binding, in contrast to the clean REST-ChIP profiles we obtain in mouse models. We therefore took an alternative, more targeted approach. We double-labeled for REST and CDK5, and for REST and GSK3 β , in human cortical brain sections from well characterized cases in the ROSMAP cohort. Quantitative analysis showed significant inverse correlations for nuclear REST and CDK5, and for nuclear REST and GSK3 β , in cortical neurons of individuals with no cognitive impairment (NCI) but with early AD pathology (Fig. 5f, g). These results in the human brain, together with compelling findings in AD mouse models, suggest that REST represses genes involved in the pathology of AD.

Reviewer #2:

Using a combination of molecular signaling, imaging, and behavioral tests on transgenic mice, and molecular signaling and imaging in human tissues, the authors demonstrate that REST serves as a neurodegeneration checkpoint with multiple molecular targets that protects against the onset of AD. The authors used a combination of genetic AD models and tested the role of REST in cognitive deficits by knockout or knockdown REST expression. By examining nuclear REST abundance in PFC (prefrontal cortex) of aging individuals with NCI or clinically diagnosed AD, the authors demonstrate that REST is induced in cognitively-intact aging AD mouse models. By inactivating REST by cKO or REST gene trap, the author further demonstrates that REST inactivation accelerates A β deposition and pathogenic tau accumulation in 3xTg mice and J20 mice. Accelerated A β deposition and tau accumulation were found in AD mice carrying a second REST loss-of-function allele. In the J20 AD mice with or without REST gene trap, the author demonstrated that loss of REST is required for A β -induced phospho-tau accumulation. Moreover, REST nuclear abundance is relevant to the cognitive decline and neurodegeneration during aging in 3xTg;cKO and J20 mice. By using ChIP-seq analysis, the authors reveal the presence of different REST targets in WT and 3xTg mice. Knockdown of REST is associated with elevated expressions of components of gamma secretase and y increased generation of A β 40 and A β 42 in vitro.

1. In a previous paper (Nature 2014), many of the same authors showed that during normal aging, REST is induced and is degraded together with pathological misfolded proteins. The conclusion is similar to that of the current manuscript that REST protects neurons from AD pathology during aging. The present paper would be significantly strengthened were the author to reveal the mechanism by which REST is induced in the early stages of AD mouse models and how REST can protect neurons.

We have added new data demonstrating a synergistic effect of the unfolded protein response (UPR) and Wnt/ β -catenin signaling in the mechanism of REST induction at early stages of pathology in the human brain and in AD mouse models (Fig. 3 and Extended Data Fig. 3). First, we determined whether Wnt/ β -catenin signaling contributed to REST induction in response to early AD pathology, as this pathway was implicated in the regulation of REST expression in our earlier paper. Examination of the aging human brain showed coordinate upregulation of nuclear REST and β -catenin in neurons of human NCI cases (cognitively intact) with early AD-type pathology relative to NCI cases with no pathology (Extended Data Fig. 3a,b). Coordinate induction of nuclear REST and β -catenin was also observed in hippocampal neurons of 3xTg mice with early pathology (Fig. 3a,b). To directly determine whether Wnt/ β -catenin signaling contributes to REST induction, we treated the 3xTg neuronal cultures with three specific inhibitors of the Wnt pathway: Dickkopf (DKK-1), an endogenous secreted antagonist of Wnt, XAV939, a tankyrase inhibitor that promotes β -catenin degradation (Huang et al., 2009), and ICG-001, a selective inhibitor of β -catenin-mediated transcription (Emami et al., 2004). Each of the three inhibitors significantly reduced REST induction in 3xTg neurons (Fig. 3c), indicating that Wnt/ β -catenin signaling contributes to REST induction. Although data supporting a role for this pathway was presented in our earlier publication, the new data strongly implicates this pathway in the context of early AD pathology in aging humans and mouse models.

An unresolved issue was how early AD pathology could lead to increased β -catenin signaling and REST induction. One of the earliest consequences of protein misfolding is activation of the unfolded protein response (UPR). Many papers have shown that the UPR is activated early in AD in response to protein misfolding, particularly tau and A β . To determine whether UPR activation affects β -catenin signaling and REST, we treated wild-type primary cortical neuronal cultures with the classic UPR activator thapsigargin (TG). Treatment with TG potentiated the nuclear translocation of β -catenin upon activation of Wnt signaling (Extended Data Fig. 3d). This was accompanied by a dramatic increase in nuclear REST (Fig. 3d). These results suggest that activation of the UPR can synergize with Wnt/ β -catenin signaling to augment the induction of REST in affected neurons.

Of the UPR pathways, PERK/eIF2 α is the first to be activated by ER stress and has been associated with both amyloid and tau pathology. Furthermore, PERK kinase activation has been shown to occur in neurons that accumulate diffuse pre-tangle tau species, but not in neurons with late-stage neurofibrillary tangles in AD (Hoozemans et al., 2009. *Am J. Pathol.* 174, 1241-1251). To determine whether the PERK/eIF2 α pathway contributes to increased REST expression, we treated primary 3xTg neurons with the PERK inhibitor GSK260414. This significantly reduced REST expression in 3xTg neurons (Fig. 3c). Taken together, these results suggest that activation of the UPR at early stages of AD pathology has a significant potentiating effect on Wnt/ β -catenin signaling and REST induction. This new insight advances our understanding of the mechanism of REST induction in aging neurons, and raises the possibility that the REST checkpoint may also be activated in other neurodegenerative disorders of protein misfolding (please see Discussion, last paragraph).

2. The authors do not sufficiently defend their conclusion that REST serves as a neurodegeneration checkpoint during aging. Most of the results are correlative. Although they use several mouse models of AD and demonstrate that loss of REST causes AD pathology and a decline in cognition, they do not examine whether overexpression of REST could reduce AD pathology or cognitive decline. Some of the MWM data are inconsistent with those obtained with other behavioral paradigms. Authors should at the least discuss this apparent discrepancy in the Discussion section.

We thank the reviewer for suggesting the overexpression approach which has turned out to be very informative. We constructed an AAV9 viral vector to transduce human REST cDNA in the mouse brain. This proved to be challenging due to insert size constraints for AAV9 vector, but was finally achieved by deletion of nonessential sequences. AAV9-hREST was introduced by stereotaxic injection into the hippocampus of 3xTg and J-20 mice. Injections with an AAV9-GFP virus served as a control. AAV9-REST overexpressed hREST mRNA by 2-3-fold relative to endogenous mouse REST in hippocampal neurons (Extended Data Fig. 11a-c). The subsequent effects on pathology were dramatic. At 10 weeks post-injection, amyloid plaque burden was markedly reduced in 16-month-old J-20 mice with a high amyloid plaque load (Fig. 9e). REST overexpression also significantly reduced pathogenic tau species in the hippocampus of 3xTg mice (Fig. 9a,b). Moreover, REST overexpression rescued the advanced tau pathology in 3xTg;GT mice with a partial REST deletion (Fig. 9c). Furthermore, REST overexpression reversed the elevated expression of REST target genes *Cdk5*, *Gsk3 β* and *Daxx* in 3xTg;GT mice (Fig. 9d). These new results strengthen the conclusion that REST activation during early stages of A β and tau accumulation can serve as a neurodegenerative checkpoint that inhibits the progression of AD-type pathology. In addition, these experiments are the first to demonstrate that a high burden of AD pathology can be reversed by modest REST overexpression, suggesting a potentially novel therapeutic approach.

The comment regarding an apparent discrepancy in behavioral paradigms is addressed in response to Specific Comment #3 below.

Specific comments:

1. Fig 1, the authors show nuclear REST levels were significantly reduced in AD cases relative to NCI cases. However, in Fig 2, their results indicate that more REST appears in the nucleus of neurons that exhibit misfolded and phosphorylated tau in 3xTg AD mouse model. Can they resolve this apparent discrepancy?

There is no discrepancy as the induction of REST is dependent on the stage. We show that REST is induced in neurons of aging individuals without cognitive decline (based on longitudinal testing) but which exhibit early stages of AD-type pathology. In contrast, REST declines significantly in cases with AD dementia that typically exhibit more advanced pathology. In the AD mouse models, REST is also induced at early stages of pathology, and we have added new data showing that it declines later just as it does in human AD. It is important to note that increased REST in the nucleus of human and mouse neurons is associated with early, diffusely distributed tau species labeled with antibody CP13 (pSer202Tau) and antibody MC1 (a conformational tau epitope) (Fig.1a, Fig. 2a and Extended Data Fig. 2a). In contrast, REST is lost from affected neurons in older 3xTg mice with more advanced tau pathology indicated by labeling with antibody PHF-1, a later stage neurofibrillary tau marker (Fig. 2b). A similar induction of REST with early A β pathology (Fig. 2c) and decline with more advanced pathology (Fig. 2d) is observed in the J20 mouse model. Hence, REST is induced by early A β and tau pathology, but not at later stages typically associated with neurofibrillary degeneration in AD. When the AD-associated decline in REST is genetically mimicked in REST-deficient 3xTg or J20 mice, tau and A β accumulation, as well as memory loss, are markedly accelerated, consistent with an inhibitory effect of REST on these processes. In contrast, REST overexpression has the opposite effect. This is the classic scenario of a biological checkpoint – a cellular response initiated by a toxic cellular insult that activates protective responses and prevents deleterious progression.

2. Fig 4, line 255, “Fig. 4e,f” should be “Fig. 4d,e”. **This has been corrected.**

And there is no J20: cHET group in the learning latency of MWM in Fig. 4d. But the authors still showed the number of entries and time spent in target area of this group in Fig. 4e.

The curve showing the learning latency of J20;cHET animals was inadvertently deleted. It is now included in Figure 8d.

3. Extended Data Fig. 7i and j: the results show that swimming speed of all three J20

mice groups is reduced compared with WT mice. However, in panel i, mice in all these three groups traveled significantly longer distances than did WT mice.

J20 mice exhibit modestly increased exploratory behavior in the open field paradigm relative to WT mice (P=0.013, Extended Data Fig. 10i). It is unclear what this reflects, which could be disinhibition, increased activity, etc. In the Morris water maze, the J20 swim speed showed a slightly reduced trend that was not statistically significant (Extended Data Fig. 10j). The important point, however, is that the three J20 mouse lines that were compared in the experiment (J20/+, J20;cHET and J20;cKO) did not differ significantly in either distance traveled in the Open Field or swim speed in the MWM (open field: J20 vs J20;cHET P=0.96; J20 vs J20;cKO P=0.75; and J20;cHET vs J20;cKO P=0.96; and swim speed in the Morris water maze: J20 vs J20;cHET P=0.99; J20 vs J20;cKO P=0.99; and J20;cHET vs J20;cKO P=0.99). Thus, differences in motor function cannot account for the decline in memory retrieval that REST-deficient J20 mice exhibit in the MWM paradigm. This conclusion is reinforced by the absence of any significant differences in the three J20 lines for time to reach a visible platform in the MWM (Extended Data Fig. 10k), which reflects a combination of motor function, stamina, motivation and visual acuity. This is now noted in the text.

4. Extended Data Fig. 9, the authors show that REST targets 555 genes in 3xTg mice (mouse brain), but 663 genes in WT mice. Thus, REST targets 115 genes exclusively in the 3xTg, but not WT, mice. To prove that not only REST, but also a host of REST target genes, are relevant to AD pathology, the authors need to provide (or at least discuss) mechanisms by which the target genes might cause neuronal death.

The data in this revised manuscript shows that REST represses the major tau kinases, as well as the subunits of gamma secretase, providing clear mechanisms for how REST represses the two major hallmarks of AD pathology. In addition, we show that REST targets well established pro-apoptotic genes, such as DAXX and p53. Furthermore, the REST ChIP-seq suggests that REST also targets genes involved in metabolism and cell cycle. We now describe the mechanistic implications of these findings in greater depth in the discussion (page 16), as suggested by the Reviewer. It is surprising that one transcription factor can target such a broad purview of pathogenic mechanisms, which may reflect a central role of REST in maintaining neural function in long-lived mammals.

Reviewer #3

Amyloid and tau pathology can be profound in brains of some aged individuals with no cognitive deficits. Investigation of the underpinning mechanisms in this specific population could provide us important insightful mechanism, herein proposed as a

mechanism of resilience. This study proposed a neurodegeneration checkpoint response manifested by REST to early pathological changes in AD that restricts further disease progression. One of these responses is to suppress development of both amyloid and tau pathology. Using a well-stratified human brain samples from the ROSMAP cohort, the authors first showed by IHC a pathology-dependent increase of nuclei REST level in PFC neurons in NCI brain samples, which was not detected in AD samples. This observation was further replicated in brains and primary neuronal cultures of aged 3xTg mice. To study the consequence of REST activation in NCI brain, they deleted REST in CA1 glutamatergic hippocampal neurons, in cortical/hip glutamatergic neurons or a het REST null allele in 3xTG with tau and amyloid or in J20 mice with amyloid only. Strikingly, they found that REST deletion substantially increased both amyloid and tau pathology, worsen cognitive deficits and increase neuronal death in both models. By performing ChIP-seq, they identified REST-targeted genes in WT and 3xTG mice. Interesting, they found some AD-pathology related genes targeted by REST, including tau kinases (such as Cdk5 and Gsk3beta) and g-secretase component PEN2. With validation of these genes in mouse models as well as in MEF cells with REST knockout or overexpression, they proposed that brain pathology induces REST activation and alter transcriptomic profile targeted by REST in AD brains, resulting in reduction of A β deposition and tau accumulation, and potentially downregulate other responses including immune responses and cell death pathways. This is a very intriguing discovery revealing an unknown mechanism how neurons response to pathology at early stage by establishing a checkpoint response to prevent further development of pathology. This finding is significant, suggesting that activating neuronal REST could lead to an effective tool for AD treatment. The experimental approaches using multiple genetically modified mouse lines are rigorous and powerful. The data provide strong support of their conclusion. Considering some concerns listed below, a revised version is recommended before acceptance for publication in this journal.

Major concerns:

1. J20 mice with massive amyloid deposition in brain have cognitive deficits in spatial learning as shown in MWM test even though REST level is increased in neurons. This is clearly different from human NCI individuals. How to justify this model in this study?

We were not clear about the time course in the last submission. REST is induced in J-20 mice during the early stage of amyloid accumulation (shown in the paper at 3 months), prior to robust deposition and plaque formation which occurs at a later age (Fig. 2c, Extended Data Fig. 2b). We now provide additional data showing that in older J20 animals with widespread amyloid deposition, REST levels are significantly reduced (Fig. 2d). This is very similar to the human brain, in which REST is induced in NCI individuals with early pathology but declines significantly in AD. Hence, the mouse model is consistent with our findings in aging humans – J20 mice show an early window of REST induction similar to that

observed cognitively intact NCI individuals at early stages of AD pathology. It is during this window that REST acts as a neurodegeneration checkpoint to restrict progression of amyloid and tau pathology and prevent cognitive decline. This is clearly suggested by the accelerated pathology (Fig. 7d; Extended Data Fig. 7e, 8) and cognitive decline (Fig. 8e) observed in REST-deficient mice, and the reversal of pathology by REST overexpression (Fig. 9e). In humans, the period of asymptomatic AD pathology, particularly A β deposition, can last for decades. Our findings suggest that REST may contribute to the resilience of the brain during this prodromal period.

2. Although the phenotype on amyloid and tau pathology by REST deletion is striking, a rescue experiment will be helpful to solidate these findings. This can be achieved by viral-mediated REST overexpression in CA1 pyramidal neurons in REST cKO mice with J20 or 3xTg background.

This was a very good suggestion. We have now performed the rescue experiment using an AAV9 viral vector to transduce hREST in 3xTg and J-20 mice. Modest REST overexpression resulted in significant attenuation of A β and tau pathology in J20 and 3xTg mice (Fig. 9a, b, e), and reversed the increase in tau pathology from partial REST deletion in 3xTg;GT mice (Fig. 9c). These findings are described in greater detail in the response to Reviewer #2, comment 2, above.

3. The sources for multiple mouse lines (where these mice were obtained) are missing in the Method, including 3xTg, J20, CA1-specific CamKII α -Cre or forebrain-specific CamKII α -Cre mice. Considering the fact that 3xTg is usually used as homozygote by this field, it is important to provide the rationale of using 3xTG het and clearly describe the breeding strategy for 3xTg with various REST mice.

The sources for mouse lines and the breeding strategies are now described in greater detail in Materials and Methods (pp. 22-23). We have used both heterozygotes and homozygotes for 3xTg in the paper. In some experiments, we used the heterozygous 3xTg because its mild phenotype was greatly augmented by REST deletion, emphasizing the important role of endogenous REST in counteracting early AD pathology.

Minor concerns:

1. Figure 1a Although the extended figure 1c shows nuclei location of REST, it is necessary to include DAPI channel in here. It seems that pTau staining in NCI early pathology samples is mainly in nuclei, which is weird. Has this been reported before? An explain or discussion of this phenotype will be helpful.

The DAPI channel is now included (Fig. 1a, and Extended Data Fig. 1d). We have performed additional double-labeling of tau and REST in NCI cases with early pathology. The accumulation of phosphorylated tau is clearly detected in neuritic

processes and neuronal somas (Fig. 1a). The observed nuclear localization of pSer202 tau was relatively weak, and has been reported previously (Eftekharzadeh et al. 2018. *Neuron* 99, 925–940).

2. Figure 1e: The amyloid pathology in 3-month-old J20 mice need to be shown.

We now show A β immunolabeling in 3-month old J20 mice. Extracellular amyloid deposition and plaques are not apparent at this early stage, but neurons are labeled weakly but specifically with A β -specific antibodies, which is not observed in WT mice (Extended Data Fig. 2b).

3. “Independent analysis did not show a significant effect of gender on the parameters described below (Extended Data 7).” It is not clear where to locate this data. Do you mean Extended Data table 7?

We apologize for his ambiguity. This data is presented in Supplementary Data Table 8: Summary of Statistical Tests.

4. Figure 2a: DAPI channel needs to be shown.

The DAPI channel is now included (Fig. 2a-d and Extended Data Fig. 2a).

5. Figure 3a,b: n=3 for REST cKO-3xTg has limited power to support the conclusion. Although the REST level is examined in WT and KO, it is still necessary to show the levels of REST in 3xTg background by IHC or western blot considering the pathology induces REST.

In addition to the 3xTg;cKO, we also show that the 3xTg;cHET (n=8 mice) also significantly accelerates tau pathology when compared to 3xTg on a normal REST background (n=15 mice) (Fig.7a, b). This is confirmed in Fig. 7b, which shows the same result with an independent set of 3xTg lines generated using a different Cre recombinase subtype that conditionally deletes REST in forebrain glutamatergic neurons (Fig. 7b: 3xTg;cKO, n=10 mice, 3xTg;cHET, n=8 mice, and 3xTg, n=8 mice).

REST levels in the 3xTg background are shown in Fig. 2a, b. The loss of REST in 3xTg;cKO mice is shown in Extended Data Fig 5d.

6. Extended data Figure 3: n=3 embryos/group, were these done in one experiment or three independent experiments? If done in one experiment, this needs to be repeated three times.

The data presented (now in Extended Data Fig. 7b,c) is from three independent experiments (in each experiment, the cortices from n=3 WT or 3xTg embryos were pooled before the cells were dissociated and plated). This methodological point is now clarified in the Extended Data Fig. 7 legend.

7. Figure 4d was not mentioned or discussed in the Result section. Figure 4f does not exist.

The data in the former Fig. 4d is now in Fig. 8d, and is described in the Results. The reference to panel f has been deleted.

8. Figure 5 and Extended figure 9: what are the color codes in these graphs?

The former Fig. 5 has been replaced with a more informative figure (now Fig. 4) showing the ChIP-seq analysis. The color codes were used to stratify different levels of statistical significance. They have been removed since numerical P-value data ($-\log_{10}P$ -values) is provided in the graphs (Fig. 4c and Extended Data Fig. 4b,c).

REVIEWER COMMENTS

Reviewer #1 (Remarks to the Author):

The authors have improved the presentation of the ChIP-seq data, however many of the caveats pointed out remain.

Differential peak enrichment analysis using statistical tools such as DiffBind or DESeq2: this wasn't performed. Given the small sample size (n=2 per condition) it is possible that such analysis would not produce significant results. GO analysis should have been performed on genes with significant differential REST enrichment. The authors have estimated statistics on ChIP-qPCR data at selected loci which doesn't consider multiple hypothesis correction. Focusing on a small set of genes strongly reduces the impact and significance of the study.

REST ChIP-seq tracks: The provided tracks show background noise at intergenic sites suggesting that the REST antibody is not very specific or that there was an issue with the IP. It is also concerning that REST is enriched at *gadph*, a housekeeping gene with known tissue-wide expression.

Genome-wide correlation between REST enrichment and gene expression: this wasn't performed. This analysis would have supported the repressive role of REST. Again, focusing on a small set of genes reduces the significance of the work.

REST ChIP-qPCR data: an IgG control should have been included.

Cell-type heterogeneity: the bulk ChIP-seq data are confounded by cell-type heterogeneity.

Reviewer #2 (Remarks to the Author):

REVIEWER COMMENTS

We thank the reviewers for their constructive comments and suggestions which have improved the paper. We believe we have addressed all the major concerns with additional experiments and analysis. The paper now provides a novel conceptual framework for the role of REST in the prevention and progression of Alzheimer's disease. The revised manuscript includes new mechanistic data on signaling pathways that activate REST in response to AD pathology, more in-depth ChIP-seq analysis, and exciting new findings on the therapeutic effects of REST overexpression in the brain. The responses to specific reviewer comments are bolded below, and major new sections of text are highlighted in the manuscript. Comments: The authors have performed experiments suggested by reviewers to reveal the molecular mechanisms to regulate REST levels during aging and effects of REST overexpression on AD pathology. In the revised manuscript, the authors added new detailed discussions to address the comments raised by reviewers and to elucidate the new data. The revisions have significantly improved the manuscript. There are still some minor issues need to be addressed.

Reviewer #2:

Using a combination of molecular signaling, imaging, and behavioral tests on transgenic mice, and

molecular signaling and imaging in human tissues, the authors demonstrate that REST serves as a neurodegeneration checkpoint with multiple molecular targets that protects against the onset of AD. The authors used a combination of genetic AD models and tested the role of REST in cognitive deficits by knockout or knockdown REST expression. By examining nuclear REST abundance in PFC (prefrontal cortex) of aging individuals with NCI or clinically diagnosed AD, the authors demonstrate that REST is induced in cognitively-intact aging AD mouse models. By inactivating REST by cKO or REST gene trap, the author further demonstrates that REST inactivation accelerates A β deposition and pathogenic tau accumulation in 3xTg mice and J20 mice. Accelerated A β deposition and tau accumulation were found in AD mice carrying a second REST loss-of-function allele. In the J20 AD mice with or without REST gene trap, the author demonstrated that loss of REST is required for A β -induced phospho-tau accumulation. Moreover, REST nuclear abundance is relevant to the cognitive decline and neurodegeneration during aging in 3xTg;cKO and J20 mice. By using CHIP-seq analysis, the authors reveal the presence of different REST targets in WT and 3xTg mice. Knockdown of REST is associated with elevated expressions of components of gamma secretase and γ increased generation of A β 40 and A β 42 in vitro

1. In a previous paper (Nature 2014), many of the same authors showed that during normal aging, REST is induced and is degraded together with pathological misfolded proteins. The conclusion is similar to that of the current manuscript that REST protects neurons from AD pathology during aging. The present paper would be significantly strengthened were the author to reveal the mechanism by which REST is induced in the early stages of AD mouse models and how REST can protect neurons.

We have added new data demonstrating a synergistic effect of the unfolded protein response (UPR) and Wnt/ β -catenin signaling in the mechanism of REST induction at early stages of pathology in the human brain and in AD mouse models (Fig. 3 and Extended Data Fig. 3). First, we determined whether Wnt/ β catenin signaling contributed to REST induction in response to early AD pathology, as this pathway was implicated in the regulation of REST expression in our earlier paper. Examination of the aging human brain showed coordinate upregulation of nuclear REST and β -catenin in neurons of human NCI cases (cognitively intact) with early AD-type pathology relative to NCI cases with no pathology (Extended Data Fig. 3a,b). Coordinate induction of nuclear REST and β catenin was also observed in hippocampal neurons of 3xTg mice with early pathology (Fig. 3a,b). To directly determine whether Wnt/ β -catenin signaling contributes to REST induction, we treated the 3xTg neuronal cultures with three specific inhibitors of the Wnt pathway: Dickkopf (DKK-1), an endogenous secreted antagonist of Wnt, XAV939, a tankyrase inhibitor that promotes β catenin degradation (Huang et al., 2009), and ICG-001, a selective inhibitor of β catenin-mediated transcription (Emami et al., 2004). Each of the three inhibitors significantly reduced REST induction in 3xTg neurons (Fig. 3c), indicating that Wnt/ β -catenin signaling contributes to REST induction. Although data supporting a role for this pathway was presented in our earlier publication, the new data strongly implicates this pathway in the context of early AD pathology in aging humans and mouse models.

Comments: The authors have used new data with inhibition of Wnt/ β -catenin signaling and demonstrated that Wnt/ β -catenin signaling contributes to REST induction in AD pathology and aging. The reviewer has no further comments with these data.

In unresolved issue was how early AD pathology could lead to increased β catenin signaling and REST induction. One of the earliest consequences of protein misfolding is activation of the unfolded protein

response (UPR). Many papers have shown that the UPR is activated early in AD in response to protein misfolding, particularly tau and A β . To determine whether UPR activation affects β -catenin signaling and REST, we treated wild-type primary cortical neuronal cultures with the classic UPR activator thapsigargin (TG). Treatment with TG potentiated the nuclear translocation of β -catenin upon activation of Wnt signaling (Extended Data Fig. 3d). This was accompanied by a dramatic increase in nuclear REST (Fig. 3d). These results suggest that activation of the UPR can synergize with Wnt/ β -catenin signaling to augment the induction of REST in affected neurons.

Comments: The authors used TG to activate UPR pathway to demonstrate UPR can synergize with Wnt/ β -catenin to induce REST. The data are solid and the conclusion is clear. In the reply to comments, the authors stated that "Treatment with TG potentiated the nuclear translocation of β -catenin upon activation of Wnt signaling (Extended Data Fig. 3d). This was accompanied by a dramatic increase in nuclear REST (Fig. 3d)". However, as shown in the Figs (3 and Extended Data) and described in the manuscript, TG itself did not significantly induce REST. TG potentiated the induction of nuclear β -catenin and REST following activation of β -catenin signaling by the drugs CHIR99021 or lithium chloride. The description in the Results section of these data are more accurate.

Of the UPR pathways, PERK/eIF2 α is the first to be activated by ER stress and has been associated with both amyloid and tau pathology. Furthermore, PERK kinase activation has been shown to occur in neurons that accumulate diffuse pre-tangle tau species, but not in neurons with late-stage neurofibrillary tangles in AD (Hoozemans et al., 2009. *Am J. Pathol.* 174, 1241-1251). To determine whether the PERK/eIF2 α pathway contributes to increased REST expression, we treated primary 3xTg neurons with the PERK inhibitor GSK260414. This significantly reduced REST expression in 3xTg neurons (Fig. 3c). Taken together, these results suggest that activation of the UPR at early stages of AD pathology has a significant potentiating effect on Wnt/ β -catenin signaling and REST induction. This new insight advances our understanding of the mechanism of REST induction in aging neurons, and raises the possibility that the REST checkpoint may also be activated in other neurodegenerative disorders of protein misfolding (please see Discussion, last paragraph).

Comments: The new data explained how activation of UPR can induce REST through PERK/eIF2 α at early stages of AD pathology. The reviewer has no further comments.

2. The authors do not sufficiently defend their conclusion that REST serves as a neurodegeneration checkpoint during aging. Most of the results are correlative. Although they use several mouse models of AD and demonstrate that loss of REST causes AD pathology and a decline in cognition, they do not examine whether overexpression of REST could reduce AD pathology or cognitive decline. Some of the MWM data are inconsistent with those obtained with other behavioral paradigms. Authors should at the least discuss this apparent discrepancy in the Discussion section.

We thank the reviewer for suggesting the overexpression approach which has turned out to be very informative. We constructed an AAV9 viral vector to transduce human REST cDNA in the mouse brain. This proved to be challenging due to insert size constraints for AAV9 vector, but was finally achieved by deletion of nonessential sequences. AAV9-hREST was introduced by stereotaxic injection into the hippocampus of 3xTg and J-20 mice. Injections with an AAV9- GFP virus served as a control. AAV9-REST overexpressed hREST mRNA by 2-3- fold relative to endogenous mouse REST in hippocampal neurons (Extended Data Fig. 11a-c). The subsequent effects on pathology were dramatic. At 10 weeks post-injection, amyloid plaque burden was markedly reduced in 16-month-old J 20 mice with a high amyloid

plaque load (Fig. 9e). REST overexpression also significantly reduced pathogenic tau species in the hippocampus of 3xTg mice (Fig. 9a,b). Moreover, REST overexpression rescued the advanced tau pathology in 3xTg;GT mice with a partial REST deletion (Fig. 9c). Furthermore, REST overexpression reversed the elevated expression of REST target genes Cdk5, Gsk3 β and Daxx in 3xTg;GT mice (Fig. 9d). These new results strengthen the conclusion that REST activation during early stages of A β and tau accumulation can serve as a neurodegenerative checkpoint that inhibits the progression of AD type pathology. In addition, these experiments are the first to demonstrate that a high burden of AD pathology can be reversed by modest REST overexpression, suggesting a potentially novel therapeutic approach.

The comment regarding an apparent discrepancy in behavioral paradigms is addressed in response to Specific Comment #3 below.

Comments: The new data using AAV to overexpress REST in the hippocampus of 3xTg and J-20 mice provide strong support of the role of REST in AD pathology and aging. REST overexpression rescued the advanced tau pathology, reversed the elevated expression of REST target genes and reduced A β plaques. The reviewer still has minor concern with the new data: In Fig 9 a and b, The CA1 structures (DAPI staining) shown in a and b are significantly different. The author needs to replace with more representative images.

Specific comments:

1. Fig 1, the authors show nuclear REST levels were significantly reduced in AD cases relative to NCI cases. However, in Fig 2, their results indicate that more REST appears in the nucleus of neurons that exhibit misfolded and phosphorylated tau in 3xTg AD mouse model. Can they resolve this apparent discrepancy?

There is no discrepancy as the induction of REST is dependent on the stage. We show that REST is induced in neurons of aging individuals without cognitive decline (based on longitudinal testing) but which exhibit early stages of AD-type pathology. In contrast, REST declines significantly in cases with AD dementia that typically exhibit more advanced pathology. In the AD mouse models, REST is also induced at early stages of pathology, and we have added new data showing that it declines later just as it does in human AD. It is important to note that increased REST in the nucleus of human and mouse neurons is associated with early, diffusely distributed tau species labeled with antibody CP13 (pSer202Tau) and antibody MC1 (a conformational tau epitope) (Fig.1a, Fig. 2a and Extended Data Fig. 2a). In contrast, REST is lost from affected neurons in older 3xTg mice with more advanced tau pathology indicated by labeling with antibody PHF-1, a later stage neurofibrillary tau marker (Fig. 2b). A similar induction of REST with early A β pathology (Fig. 2c) and decline with more advanced pathology (Fig. 2d) is observed in the J20 mouse model. Hence, REST is induced by early A β and tau pathology, but not at later stages typically associated with neurofibrillary degeneration in AD. When the AD-associated decline in REST is genetically mimicked in REST-deficient 3xTg or J20 mice, tau and A β accumulation, as well as memory loss, are markedly accelerated, consistent with an inhibitory effect of REST on these processes. In contrast, REST overexpression has the opposite effect. This is the classic scenario of a biological checkpoint – a cellular response initiated by a toxic cellular insult that activates protective responses and prevents deleterious progression.

Comments: The authors have addressed my comments and the reviewer has no further comments.

2. Fig 4, line 255, “Fig. 4e,f” should be “Fig. 4d,e”. This has been corrected. And there is no J20: cHET

group in the learning latency of MWM in Fig. 4d. But the authors still showed the number of entries and time spent in target area of this group in Fig. 4e.

The curve showing the learning latency of J20;cHET animals was inadvertently deleted. It is now included in Figure 8d.

Comments: The authors have corrected the information pointed out by the reviewer in the revised version.

3. Extended Data Fig. 7i and j: the results show that swimming speed of all three J20 mice groups is reduced compared with WT mice. However, in panel i, mice in all these three groups traveled significantly longer distances than did WT mice.

J20 mice exhibit modestly increased exploratory behavior in the open field paradigm relative to WT mice ($P=0.013$, Extended Data Fig. 10i). It is unclear what this reflects, which could be disinhibition, increased activity, etc. In the Morris water maze, the J20 swim speed showed a slightly reduced trend that was not statistically significant (Extended Data Fig. 10j). The important point, however, is that the three J20 mouse lines that were compared in the experiment (J20/+, J20;cHET and J20;cKO) did not differ significantly in either distance traveled in the Open Field or swim speed in the MWM (open field: J20 vs J20;cHET $P=0.96$; J20 vs J20;cKO $P=0.75$; and J20;cHET vs J20;cKO $P=0.96$; and swim speed in the Morris water maze: J20 vs J20;cHET $P=0.99$; J20 vs J20;cKO $P=0.99$; and J20;cHET vs J20;cKO $P=0.99$). Thus, differences in motor function cannot account for the decline in memory retrieval that REST-deficient J20 mice exhibit in the MWM paradigm. This conclusion is reinforced by the absence of any significant differences in the three J20 lines for time to reach a visible platform in the MWM (Extended Data Fig. 10k), which reflects a combination of motor function, stamina, motivation and visual acuity. This is now noted in the text.

Comments: The Data from a visible platform in the MWM is strong evidence to support that for the decline in memory of animal models is not caused by impaired motor function. The authors also provide adequate explanation about the behavioral results in Extended Data Fig. 7i and j. The reviewer has no further comment.

4. Extended Data Fig. 9, the authors show that REST targets 555 genes in 3xTg mice (mouse brain), but 663 genes in WT mice. Thus, REST targets 115 genes exclusively in the 3xTg, but not WT, mice. To prove that not only REST, but also a host of REST target genes, are relevant to AD pathology, the authors need to provide (or at least discuss) mechanisms by which the target genes might cause neuronal death. The data in this revised manuscript shows that REST represses the major tau kinases, as well as the subunits of gamma secretase, providing clear mechanisms for how REST represses the two major hallmarks of AD pathology. In addition, we show that REST targets well established pro-apoptotic genes, such as DAXX and p53. Furthermore, the REST ChIP-seq suggests that REST also targets genes involved in metabolism and cell cycle. We now describe the mechanistic implications of these findings in greater depth in the discussion (page 16), as suggested by the Reviewer. It is surprising that one transcription factor can target such a broad purview of pathogenic mechanisms, which may reflect a central role of REST in maintaining neural function in long-lived mammals.

Comments: The authors added new discussion on that REST may potentially affect neuronal death during aging and AD pathology through regulating genes in cell death (p53), cell cycle and metabolism. The reviewer has no further comment.

Reviewer #3 (Remarks to the Author):

The authors have addressed my previous concerns. But I want to point out an issue with statistical analysis. Many analyses involving two variables were done by One-way Anova, which is not correct method. Two way Anova should be used. For example, Figure 1b, NCI vs AD and with or without pathology are two variables, Figure 8b, c, e with 3xTg or WT vs cHET variable, the authors used One way anova with Tukey's post-hot test in these tests.

Response to reviewer comments:

We thank the reviewers for constructive comments. We believe that the remaining concerns have been addressed with additional experiments, particularly as regards the ChIP-seq analysis. The additional ChIP-seq data solidifies the earlier findings and provides a sound statistical basis for the conclusions. The responses to specific reviewer comments are bolded below, and new sections of text are highlighted in the manuscript.

REVIEWER COMMENTS

Reviewer 1

The authors have improved the presentation of the ChIP-seq data, however many of the caveats pointed out remain. Differential peak enrichment analysis using statistical tools such as DiffBind or DESeq2: this wasn't performed. Given the small sample size (n=2 per condition) it is possible that such analysis would not produce significant results. GO analysis should have been performed on genes with significant differential REST enrichment.

We have increased the REST ChIP-seq sample size to n=4 per condition (WT and 3xTg). A differential peak enrichment analysis was then performed using the *DiffBind* software, as suggested by the reviewer. As shown in the revised manuscript (Figure 4, Extended Fig. 4 and Extended Data Tables 2, 3, 4 and 5), we have identified 507 peaks with significantly increased REST binding in the 3xTg cortex (FDR<0.05). Importantly, the REST targets that we previously validated by ChIP-qPCR (*gsk3 β* , *cdk5*, *daxx*), and by REST loss- and gain-of-function, immunofluorescence, and Western blotting, were also validated in this expanded analysis with a statistically-significant increase in REST binding in 3xTg mice. GO enrichment analysis of all genes with significantly increased REST binding in 3xTg mice showed significant enrichment of REST targets in GO groups corresponding to cellular metabolism, synapse organization and transport, cell cycle, stress responses and apoptotic cell death. In summary, the new ChIP-seq data further strengthens our conclusion that REST is induced in the setting of early AD pathology and targets categories of genes that are highly relevant to AD pathogenesis (Discussion, pages 15-17). The ChIP-seq tracks for WT and 3xTg (mean of 4 replicates/group) can be accessed online at https://genome.ucsc.edu/s/zhenkai/mm10_REST_ChIPseq_2.

The authors have estimated statistics on ChIP-qPCR data at selected loci which doesn't consider multiple hypothesis correction. Focusing on a small set of genes strongly reduces the impact and significance of the study.

As described above, the statistical power of ChIP-seq has been increased and *DiffBind* was applied to the data which incorporates multiple test correction (FDR<0.05). Rigorous statistical analysis with multiple test correction was also applied to the ChIP-qPCR data as follows: 1) for each gene, one-way ANOVA was applied with correction for multiple comparisons (e.g. REST binding in the peak region in WT vs 3xTg; comparison of REST antibody binding vs. non-specific IgG control, for either WT or 3xTg; comparison of REST binding in the peak region vs. 10 kb downstream, for either WT or 3xTg); 2). We further applied Bonferroni correction to all adjusted P-values after the one-way ANOVA to account for the simultaneous analysis of 6 genomic loci (*gsk3β*, *cdk5*, *daxx*, *tle3*, *cacng2*, *untr6*). The resulting adjusted P-values remain highly significant ($p < 0.001$ and **** $p < 0.0001$; see Fig. 4d and Extended Data Table 8), indicating that REST binding to these genes is significantly stronger in 3xTg vs WT. Moreover, we tested 2 independent cohorts of WT and 3xTg mice (age 5 and 11 months). Differential REST binding in 3xTg vs WT was highly significant at both ages, suggesting a prolonged and robust effect.**

REST ChIP-seq tracks: The provided tracks show background noise at intergenic sites suggesting that the REST antibody is not very specific or that there was an issue with the IP. It is also concerning that REST is enriched at *gapdh*, a housekeeping gene with known tissue-wide expression.

It is typical for there to be some intergenic background signal in ChIP-seq. In order to help distinguish significant peaks from background signal, we have now included the callpeak bed files together with the new tracks, which highlights robust REST binding at statistically significant peak regions in all replicates. With respect to *gapdh*, low level signal can be discerned in two of the eight replicates, however the application of MACS2 indicates that these are not statistically significant peaks.

Genome-wide correlation between REST enrichment and gene expression: this wasn't performed. This analysis would have supported the repressive role of REST. Again, focusing on a small set of genes reduces the significance of the work.

As we show, REST is selectively induced in specific neuronal subpopulations in the AD patients and in two AD mouse models at an early stage of pathology. Bulk tissue RNA seq is not a very good assessment of the consequence of this selective increase in REST binding due to background transcription in unaffected cell types. However, bulk ChIP-seq resolves increased binding in 3xTg quite effectively because REST binding to many of these genes is at background levels in WT mice. The Reviewer suggests that our focus on key pathology-related REST targets without a more comprehensive analysis of the repressive role of REST on

gene expression “reduces the significance of the work”. However, 25 years of investigation of REST has established that it is a transcriptional repressor. As such, reiterating this well-known aspect on a genome-wide level may not significantly impact the paper's overall significance. The significance and novelty of this paper is the demonstration that REST is induced at the early stages of AD pathology and targets multiple pathogenic pathways, including major tau kinases that mediate neurofibrillary pathology and gamma secretase genes that mediate amyloid pathology. This provides a novel conceptual framework underlying the asymptomatic period of early AD pathology, which can last for decades. Furthermore, the REST gain-of-function experiments *in vivo* suggest a possible new therapeutic approach. We have taken the Reviewer's comments into consideration and have made appropriate revisions to better convey the significance of our findings (see Discussion, pages 15-17).

REST CHIP-qPCR data: an IgG control should have been included.

We have included an IgG control for the ChIP-qPCR analysis. In contrast to anti-REST IgG, the non-specific IgG antibody does not bind to the peak regions of *gsk3 β* , *cdk5* and *daxx* (Fig. 4d).

Cell-type heterogeneity: the bulk ChIP-seq data are confounded by cell-type heterogeneity.

Our REST immunolabeling (Fig. 2) shows that REST is induced preferentially in phospho tau-positive neurons in the 3xTg cortex. As such, we expect that REST binding will be enriched in this neuronal population relative to phospho tau-negative neurons and glial cells. ChIP-seq using purified cellular populations, or individual cells, might provide additional cell type information, but it is technically challenging given the amount of tissue that would be required, and is unlikely to add to the major conclusions of the paper. We plan to perform higher resolution single cell analysis using ATAC seq, but this is beyond the scope of the present manuscript.

Reviewer 2

Comments: The authors have performed experiments suggested by reviewers to reveal the molecular mechanisms to regulate REST levels during aging and effects of REST overexpression on AD pathology. In the revised manuscript, the authors added new detailed discussions to address the comments raised by reviewers and to elucidate the new data. The revisions have significantly improved the manuscript. There are still some minor issues need to be addressed.

Comments: The authors have used new data with inhibition of Wnt/ β -catenin signaling and demonstrated that Wnt/ β -catenin signaling contributes to REST induction in AD pathology and aging. The reviewer has no further comments with these data.

Comments: The authors used TG to activate UPR pathway to demonstrate UPR can synergize with Wnt/ β -catenin to induce REST. The data are solid and the conclusion is clear. In the reply to comments, the authors stated that "Treatment with TG potentiated the nuclear translocation of β -catenin upon activation of Wnt signaling (Extended Data Fig. 3d). This was accompanied by a dramatic increase in nuclear REST (Fig. 3d)". However, as shown in the Figs (3 and Extended Data) and described in the manuscript, TG itself did not significantly induce REST. TG potentiated the induction of nuclear β -catenin and REST following activation of β -catenin signaling by the drugs CHIR99021 or lithium chloride. The description in the Results section of these data are more accurate.

We apologize for the lack of clarity in our reply to the reviewer's comments. As the reviewer notes, the data is accurately described in the manuscript.

Comments: The new data explained how activation of UPR can induce REST through PERK/eIF2 α at early stages of AD pathology. The reviewer has no further comments.

Comments: The new data using AAV to overexpress REST in the hippocampus of 3xTg and J-20 mice provide strong support of the role of REST in AD pathology and aging. REST overexpression rescued the advanced tau pathology, reversed the elevated expression of REST target genes and reduced A β plaques. The reviewer still has minor concern with the new data: In Fig 9 a and b, The CA1 structures (DAPI staining) shown in a and b are significantly different. The author needs to replace with more representative images.

We have provided more representative images of the CA1 sector of the hippocampus in Fig. 9a, b.

Comments: The authors have addressed my comments and the reviewer has no further comments.

Comments: The authors have corrected the information pointed out by the reviewer in the revised version.

Comments: The Data from a visible platform in the MWM is strong evidence to support that for the decline in memory of animal models is not caused by impaired motor function. The authors also provide adequate explanation about the behavioral results in Extended Data Fig. 7i and j. The reviewer has no further comment.

Comments: The authors added new discussion on that REST may potentially affect

neuronal death during aging and AD pathology through regulating genes in cell death (p53), cell cycle and metabolism. The reviewer has no further comment.

Reviewer 3

The authors have addressed my previous concerns. But I want to point out an issue with statistical analysis. Many analyses involving two variables were done by One-way Anova, which is not correct method. Two way Anova should be used. For example, Figure 1b, NCI vs AD and with or without pathology are two variables, Figure 8b, c, e with 3xTg or WT vs cHET variable, the authors used One way anova with Tukey's post-hot test in these tests.

We have provided two-way ANOVA analyses as suggested by the reviewer, for Fig. 1b, Fig. 8a, Fig. 8c,e, Extended Data Fig. 1a, Extended Data Fig. 9c, Extended Data Fig. 10a-c, and Extended Data Fig. 10i-k. The analyses and the P-values are provided in the Extended Data Table 8. We indicated in the corresponding figure legends if the two-way ANOVA analyses uncovered statistically-significant interactions between the variables tested. If a two-way ANOVA analysis did not uncover a significant interaction between the 2 variables tested, we conducted a one-way ANOVA analysis and presented the results of both tests in the Extended Data Table 8.

REVIEWERS' COMMENTS

Reviewer #1 (Remarks to the Author):

The authors have improved the REST ChIP-seq data analysis by increasing the sample size to N=4 per group. Differential peak enrichment analysis revealed genomic sites with increased REST binding in the 3xTg mice and highlighted relevant GO terms. The analysis also confirmed the previous ChIP-qPCR results.

The authors didn't clarify whether peaks with loss in REST were also detected in the 3xTg mice, and the implicated genes. This should be added to the manuscript.

The authors have addressed the main concerns and I have no further comments apart from the one above.

Reviewer #2 (Remarks to the Author):

I only had two minor comments last time. The authors have answered our comment with new results and corrected the inaccurate descriptions as shown below. The new results (images) are solid and support the authors' conclusion. I do not have further comments.

REVIEWER COMMENTS

We thank the reviewers for their careful consideration of the paper and comments which have improved the manuscript.

Reviewer #1:

The authors have improved the REST ChIP-seq data analysis by increasing the sample size to N=4 per group. Differential peak enrichment analysis revealed genomic sites with increased REST binding in the 3xTg mice and highlighted relevant GO terms. The analysis also confirmed the previous ChIP-qPCR results.

The authors didn't clarify whether peaks with loss in REST were also detected in the 3xTg mice, and the implicated genes. This should be added to the manuscript.

The authors have addressed the main concerns and I have no further comments apart from the one above.

We have not detected peaks with significantly reduced REST binding in 3xTg relative to WT mice. This information has now been added to the “Results” section (page 7, second paragraph): “Analysis of differentially bound genes in WT and 3xTg identified 507 peaks (635 genes) with significantly higher REST occupancy, and no peaks with significantly decreased REST occupancy, in the 3xTg cortex (Supplementary Data 5).”

Reviewer #2 (Remarks to the Author):

I only had two minor comments last time. The authors have answered our comment with new results and corrected the inaccurate descriptions as shown below. The new results (images) are solid and support the authors' conclusion. I do not have further comments.

We thank the Reviewer for comments and suggestions.